# cuRegOT: A GPU-Accelerated Solver for Entropic-Regularized Optimal Transport

Yixuan Qiu [1]

## Abstract

Optimal transport (OT) has emerged as a fundamental tool in modern machine learning, yet its computational cost remains a significant bottleneck for large-scale applications. While harnessing the massive parallelism of modern GPU hardware is critical for efficiency, the *de facto* standard Sinkhorn algorithm, despite its ease of parallelization, often suffers from slow convergence in challenging problems. More recently, the sparse-plus-low-rank quasi-Newton method offers a balance between convergence rate and per-iteration complexity; however, its efficiency on GPUs is severely hindered by the serial nature of sparse matrix symbolic analysis and irregular memory access patterns. To bridge this gap, we present cuRegOT, a high-performance GPU solver tailored for entropic-regularized OT. We introduce a suite of algorithmic and architectural optimizations, including an amortized symbolic analysis strategy to mitigate CPU bottlenecks, an asynchronous Sinkhorn iterates generation mechanism, and a fused kernel for bandwidth-efficient gradient evaluation. These strategies are backed by rigorous theoretical guarantees ensuring algorithmic convergence. Extensive numerical experiments demonstrate that cuRegOT achieves significant speedups over state-of-the-art GPU-based solvers across a variety of benchmark tasks.

## 1 Introduction

Optimal transport (OT) provides a principled way to compare probability measures by seeking the minimum cost of coupling two distributions under a prescribed ground cost

---

[1]School of Statistics and Data Science & Institute of Big Data Research, Shanghai University of Finance and Economics, Shanghai, China. Correspondence to: Yixuan Qiu <qiuyixuan@sufe.edu.cn>.

*Proceedings of the 43$^{rd}$ International Conference on Machine Learning*, Seoul, South Korea. PMLR 306, 2026. Copyright 2026 by the author(s).

(Villani et al., 2009). The resulting Wasserstein distance endows the space of probability measures with a rich geometry, and has found widespread success in statistical machine learning, computer vision, natural language processing, and modern data science (Peyré & Cuturi, 2019).

In machine learning, OT has been widely adopted as a loss function or regularizer for tasks that require aligning probability distributions. Representative examples include unsupervised domain adaptation (Courty et al., 2017), generative modeling via Wasserstein GAN (Arjovsky et al., 2017), computer graphics (Solomon et al., 2015), and single-cell biology (Schiebinger et al., 2019), among many others. These applications often require solving OT problems repeatedly, making the computational efficiency and hardware scalability of OT solvers a prioritized concern.

The discrete OT problem is a linear programming problem:

$$\min_{P \in \Pi(a,b)} \langle P, M \rangle, \tag{1}$$

where $M \in \mathbb{R}^{n \times m}$ is a given cost matrix, $\Pi(a,b) = \{P \in \mathbb{R}^{n \times m} : P\mathbf{1}_m = a, P^T\mathbf{1}_n = b, P \geq 0\}$, $a$ and $b$ are two probability vectors satisfying $a > 0$, $b > 0$, and $\sum_{i=1}^{n} a_i = \sum_{j=1}^{m} b_j = 1$, and all inequality signs applied to vectors and matrices are elementwise.

Despite its elegance, solving (1) at scale is challenging. Standard linear programming solvers typically incur a supercubic complexity of $O(n^3 \log(n))$ for $n \approx m$ (Pele & Werman, 2009). This computational burden becomes prohibitive when dealing with high-dimensional data or large support sizes (e.g., $n, m > 10^4$), necessitating the development of more efficient approximation schemes.

To overcome the computational difficulty of solving (1), Cuturi (2013) proposes the entropic-regularized OT problem as an approximation to the original OT problem, which adds an entropic regularization to the objective function of (1):

$$\min_{P \in \Pi(a,b)} \langle P, M \rangle - \eta \cdot h(P), \tag{2}$$

where $h(P) = \sum_{i,j} P_{ij}(1 - \log(P_{ij}))$ is the entropy term. At first glance, problem (2) is no simpler than (1), but its

benefit is clear by studying the dual problem of (2):

$$\max_{\alpha \in \mathbb{R}^n, \beta \in \mathbb{R}^m} \mathcal{L}(\alpha, \beta), \tag{3}$$

$$\mathcal{L}(\alpha, \beta) = -\eta \sum_{i=1}^{n} \sum_{j=1}^{m} \exp\{\eta^{-1}(\alpha_i + \beta_j - M_{ij})\}$$
$$+ \alpha^T a + \beta^T b.$$

Clearly, (3) is a concave, smooth, and unconstrained maximization problem, which allows for many first- and second-order optimization techniques. Let $T^*$ and $(\alpha^*, \beta^*)$ be the primal and dual optima of the problems (2) and (3), respectively. Then they are connected by the relation $T_{ij}^* = \exp\{(\alpha_i^* + \beta_j^* - M_{ij})/\eta\}$.

Cuturi (2013) uses the well-known Sinkhorn algorithm (Yule, 1912; Sinkhorn, 1964) to solve (2), which is equivalent to applying the block coordinate ascent method to the dual problem (3). One major advantage of the Sinkhorn algorithm is that it can be highly parallelized, thus suitable for GPU implementation. This has led to the development of several widely-used GPU-based OT solvers. For instance, POT (Python optimal transport, Flamary et al., 2021) can use the CuPy package (Okuta et al., 2017) as a backend to support GPU computing, while OTT-JAX (Cuturi et al., 2022) leverages the JAX framework to offer just-in-time compilation on accelerators.

However, empirical results show that the Sinkhorn algorithm may demonstrate slow convergence on challenging problems. More recently, there is an increasing interest in applying second-order or quasi-Newton method to solving (3). As a preliminary, note that $(\alpha, \beta)$ has one redundant degree of freedom, as $\mathcal{L}(\alpha, \beta) \equiv \mathcal{L}(\alpha + c\mathbf{1}_n, \beta - c\mathbf{1}_m)$ for all $c \in \mathbb{R}$. Therefore, by fixing $\beta_m = 0$ and defining the free variable $x = (\alpha, \beta_{-m})$, where $\beta_{-m} = (\beta_1, \ldots, \beta_{m-1})^T$, our goal reduces to the smooth convex optimization problem

$$\min_{x \in \mathbb{R}^{n+m-1}} f(x), \quad f(x) = -\mathcal{L}(\alpha, \beta). \tag{4}$$

Several advanced Newton-type methods have been proposed to solve (4). For instance, the SNS (Tang et al., 2024) and SSNS (Tang & Qiu, 2024) algorithms use second-order search directions to accelerate convergence, and exploit the approximate sparsity in the Hessian matrices to control the per-iteration cost. Additionally, the sparse-plus-low-rank algorithm (SPLR, Wang & Qiu, 2025) further incorporates the specific structure of the Hessian matrices in entropic-regularized OT, and develops a quasi-Newton method that improves robustness when the transport plan is dense.

While these Newton-type methods demonstrate promising convergence speeds compared to the Sinkhorn algorithm, their practical efficiency on GPUs remains an open challenge. The primary bottleneck lies in solving large sparse

linear systems required for Newton steps. Unlike the dense matrix-vector operations in Sinkhorn, sparse factorizations typically rely on symbolic analysis and reordering steps that are inherently sequential and difficult to parallelize on SIMT (single instruction, multiple threads) architectures. Consequently, existing implementations of these advanced algorithms are often confined to CPUs or suffer from severe GPU under-utilization.

To address this gap, this article presents the cuRegOT library (CUDA-accelerated regularized optimal transport), a high-performance GPU solver tailored for entropic-regularized OT. cuRegOT adds important improvements to the SPLR algorithm, and is programmed on the CUDA parallel computing platform. Specifically, we focus on GPU-oriented algorithm and system designs to mitigate the bottlenecks in solving sparse Newton-type linear systems, including: (1) amortizing symbolic analysis by reusing sparsity patterns across multiple iterations; (2) overlapping CPU-side symbolic analysis with auxiliary GPU-side candidate iterate updates to better utilize accelerator resources; and (3) developing efficient fused CUDA kernels for gradient evaluation. A preview of the cuRegOT execution pipeline is illustrated in Figure 1, with details elaborated in subsequent sections. The source code of cuRegOT is available at https://github.com/yixuan/regot-cuda.

## 2 Related Work

**Entropic-regularized OT** Entropic regularization turns discrete OT into a strictly convex problem whose dual is smooth and unconstrained, enabling scalable iterative solvers (Cuturi, 2013; Peyré & Cuturi, 2019). The dominant approach is Sinkhorn matrix scaling (Sinkhorn, 1964), together with stabilization and $\varepsilon$-scaling heuristics for small regularization (Schmitzer, 2019). Recent works revisit complexity and acceleration of Sinkhorn-type schemes, e.g., greedy coordinate variants and improved analyses (Altschuler et al., 2017; Lin et al., 2022). Despite these progresses, widely-used GPU solvers for entropic OT still largely center on Sinkhorn-style iterations.

**GPU-based OT solvers** Because Sinkhorn iterations are dominated by dense kernel evaluations and elementwise operations, most practical GPU OT solvers focus on Sinkhorn and its variants, including POT (Flamary et al., 2021) and OTT-JAX (Cuturi et al., 2022). Beyond entropic-regularized OT, PDOT (Lu & Yang, 2024) provides a GPU solver aimed at high-accuracy solutions for the *unregularized* OT linear program. In addition, Douglas–Rachford splitting has been adapted to deliver an efficient GPU implementation for a range of *non-entropic* regularizers (Lindbäck et al., 2023), explicitly contrasting with entropic regularization.

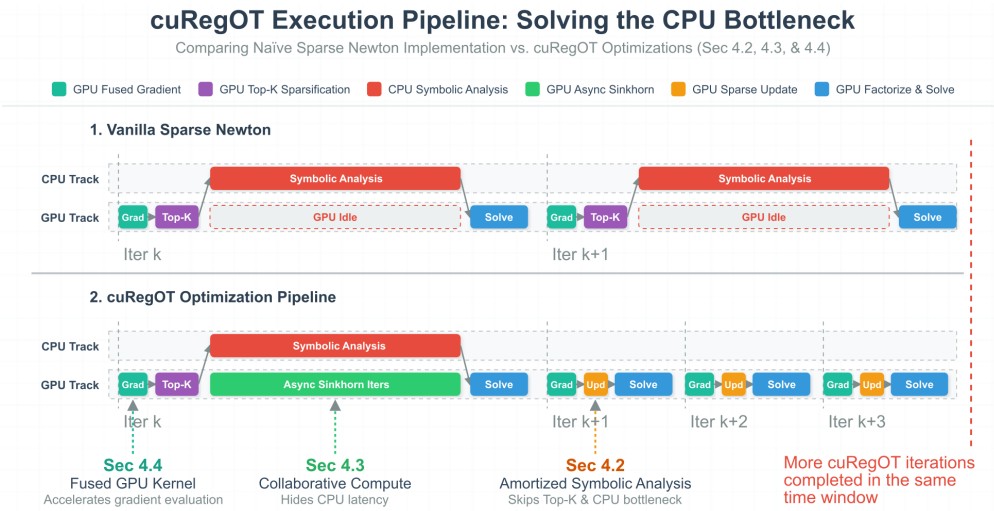

*Figure 1.* An overview of the cuRegOT pipeline.

**Quasi-Newton methods** Quasi-Newton methods approximate curvature using gradient information and are widely used in large-scale optimization due to their strong practical performance (Nocedal & Wright, 2006). In entropic-regularized OT, recent Newton-type solvers exploit the structure in the dual curvature, notably through Hessian sparsification and low-rank approximation (Tang et al., 2024; Tang & Qiu, 2024; Wang & Qiu, 2025; Ouyang et al., 2026). However, realizing these methods efficiently on GPUs is challenging because each outer step typically requires solving large sparse linear systems. This motivates our GPU-oriented algorithm and system designs that attempt to overcome this difficulty.

## 3 Motivation and Preliminaries

### 3.1 The SPLR Algorithm

Our proposed solver, cuRegOT, is based on the SPLR algorithm for entropic-regularized OT, with special designs tailored for modern GPU hardware. In this section we provide a brief and self-contained introduction to the SPLR algorithm. Recall that our goal is to solve the smooth convex optimization problem (4), and SPLR can be viewed as a quasi-Newton method, in the sense that it uses both the gradient $\nabla f(x)$ and an approximation to the Hessian $\nabla^2 f(x)$ to generate a sequence of iterates $\{x_k\}$.

It can be shown that the gradient function and the Hessian matrix have closed-form expressions:

$$\nabla f(x) = \begin{bmatrix} T\mathbf{1}_m - a \\ T_{-m}^T \mathbf{1}_n - b_{-m} \end{bmatrix},$$

$$\nabla^2 f(x) = \eta^{-1} \begin{bmatrix} \mathbf{diag}(T\mathbf{1}_m) & T_{-m} \\ T_{-m}^T & \mathbf{diag}(T_{-m}^T \mathbf{1}_n) \end{bmatrix},$$

where $T = T(x)$ is an $n \times m$ matrix with elements $T_{ij} = \exp\{(\alpha_i + \beta_j - M_{ij})/\eta\}$, given the variable $x = (\alpha, \beta_{-m})$, and $T_{-m}$ means removing the $m$-th column from $T$.

The SPLR algorithm generates iterates using the update rule $x^+ = x - \gamma B^{-1} g$, where $x$ and $g = \nabla f(x)$ represent the current iterate and gradient, respectively, $x^+$ is the next iterate, $\gamma$ is a step size that may vary with the iteration, and the matrix $B$ is an approximation to the current Hessian matrix, $B \approx H = \nabla^2 f(x)$. $B$ is chosen such that it contains useful information about $H$ and meanwhile admits faster computation of $B^{-1} g$ than $H^{-1} g$. Unlike the well-known BFGS method that constructs $B$ using purely diagonal and low-rank matrices, the SPLR algorithm defines $B$ as

$$B = H_\Omega + (\xi u u^T + \zeta v v^T) + \tau I, \tag{5}$$

where $H_\Omega$ is a sparse matrix, $(\xi u u^T + \zeta v v^T)$ is a rank-two matrix, and $\tau I$ is diagonal. The rank-two term inherits the structure of the BFGS rule, and is determined by the following procedure. Let $x^-$ and $g^- = \nabla f(x^-)$ be the previous iterate and gradient, respectively, and define $y^- = g - g^-$ and $s^- = x - x^-$. Then we set

$$u = y^-, \qquad v = (H_\Omega + \tau I)s^-,$$
$$\xi = \frac{1}{(y^-)^T s^-}, \quad \zeta = -\frac{1}{v^T s^-}. \tag{6}$$

The diagonal term can be simply set to $\tau = \min\{\tau_{\max}, \|g\|\}$ for some fixed constant $\tau_{\max} > 0$.

The sparse term $H_\Omega$ is the core component of the SPLR algorithm, which has the following structure:

$$H_\Omega = \eta^{-1} \begin{bmatrix} \mathbf{diag}(T\mathbf{1}_m) & T_{-m}^\Omega \\ (T_{-m}^\Omega)^T & \mathbf{diag}(T_{-m}^T \mathbf{1}_n) \end{bmatrix}, \tag{7}$$

where $T^{\Omega}_{-m}$ is an $n \times (m-1)$ matrix with entries

$$(T^{\Omega}_{-m})_{ij} = \begin{cases} T_{ij}, & (i,j) \in \Omega, \\ 0, & (i,j) \notin \Omega, \end{cases}$$

and $\Omega$ is a subset of the matrix coordinates, $\Omega \subseteq \{(i,j) : 1 \le i \le n, 1 \le j \le m-1\}$. Since $H_{\Omega}$ is primarily determined by the index set $\Omega$, we call $\Omega$ a *sparsification scheme* for clarity. For theoretical analysis, we assume that $\Omega$ contains a minimum set of indices: $\Omega \supseteq \Omega^* = \{(i,j) : i = 1 \text{ or } j = 1, 1 \le i \le n, 1 \le j \le m-1\}$.

Finally, the step size $\gamma > 0$ is typically determined by a line search procedure (Moré & Thuente, 1994) that guarantees the Wolfe conditions:

$$\begin{aligned} f(x^+) &\le f(x) + c_1 \gamma g^T d, \\ [\nabla f(x^+)]^T d &\ge c_2 g^T d, \end{aligned} \quad (8)$$

where $d = -B^{-1}g$, and $0 < c_1 < 1/2$ and $c_1 < c_2 < 1$ are pre-specified constants. The overall structure of the SPLR algorithm, after omitting some minor details such as the iteration loop and the convergence test, is summarized in Algorithm 1. In our implementation of cuRegOT, the sparsification scheme $\Omega$ is based on a simple top-$k$ rule: we select the largest $k$ entries of $T$, forming $T^{\Omega}$ by zeroing out all other entries. Then $H_{\Omega}$ is assembled using formula (7). Readers are referred to Wang & Qiu (2025) for more details of choosing $\Omega$ in each iteration.

---

**Algorithm 1** Overview of the SPLR algorithm for solving entropic-regularized OT.

**Input:** Previous iterate and gradient $(x^-, g^-)$, current iterate $x$, constant $\tau_{\max} > 0$
**Output:** Next iterate $x^+$
1: Compute $T = T(x)$, $g = \nabla f(x)$, $H = \nabla^2 f(x)$
2: Determine $\Omega$ and compute $H_{\Omega}$
3: Compute $s^- = x - x^-$ and $y^- = g - g^-$
4: Compute $\xi, \zeta, u, v$ according to (6)
5: Let $R = \begin{cases} \xi u u^T + \zeta v v^T, & \text{if } (y^-)^T s^- > 10^{-6} \|y^-\|^2 \\ O, & \text{otherwise} \end{cases}$
6: Set $\tau = \min\{\tau_{\max}, \|g\|\}$
7: Compute $d = -B^{-1}g$, where $B = H_{\Omega} + R + \tau I$
8: **return** $x^+ = x + \gamma d$ with $\gamma$ selected by line search

---

### 3.2 Challenges on GPU Implementation

The SPLR algorithm has achieved promising computational efficiency on the CPU-based implementation, compared with well-known reference algorithms including the Sinkhorn algorithm. However, it is non-trivial to adapt it to GPU due to its major computational bottleneck on computing the quasi-Newton search direction. Specifically, one of the most critical part of the algorithm is computing the

sparse Cholesky decomposition of the sparsified Hessian matrix $H_{\Omega}$, which mainly consists of three steps:

1. **Symbolic analysis**: analyzing the sparsity pattern of $H_{\Omega}$, and determining the reordering method for $H_{\Omega}$. Reordering the rows and columns of $H_{\Omega}$ is critical to achieve a sparse decomposition result, in the sense that $L$ may be sparser than $L_0$ with $P^T H_{\Omega} P = LL^T$ and $H_{\Omega} = L_0 L_0^T$, for some permutation matrix $P$.
2. **Factorization**: computing the sparse lower-triangular $L$ matrix given $H_{\Omega}$ and the reordering method $P$.
3. **Solving**: obtaining $H_{\Omega}^{-1} g = P(L^T)^{-1} L^{-1} P^T g$ by computing the forward substitution $L^{-1}u$ and the backward substitution $(L^T)^{-1}v$ for some vectors $u$ and $v$.

For most existing GPU-based sparse Cholesky solvers, the factorization and solving steps can well support parallel computing, whereas the symbolic analysis part primarily runs on CPU, and typically takes up a significant amount of computing time. This fact becomes the major obstacle to adapting the SPLR algorithm to efficient GPU computing.

## 4 Method: The cuRegOT Solver

### 4.1 Overview

To overcome the issues introduced in Section 3.2 and to facilitate efficient OT solvers, we propose three algorithm and system designs that are tailored for modern GPU. Below we provide an overview of the methods we develop, with details of these designs elaborated in subsequent sections:

- We show with theoretical guarantees that the CPU computing bottleneck can be amortized, in the sense that the symbolic analysis result can be reused by multiple iterations. On average, the cost resulted from pure CPU computing can be reduced by a large factor.
- We design an asynchronous algorithm that allow GPU to compute useful information when CPU is working on the symbolic analysis. This additional information can potentially accelerate the convergence of the algorithm, thus reducing the total runtime.
- We design a fused CUDA kernel for efficient gradient evaluation, aiming at reducing overhead and memory IO.

### 4.2 Amortized Symbolic Analysis of Sparsity Pattern

The standard SPLR algorithm re-evaluates the sparsity pattern of $H_{\Omega}$ at every iteration, which depends on the value of the current iterate $x$. However, performing symbolic analysis and matrix reordering (e.g., via nested dissection) to minimize fill-in during sparse Cholesky decomposition is a highly serial process, typically executed on the CPU. This creates a significant synchronization bottleneck, leaving the GPU idle between numerical steps.

We observe that the sparsity pattern, typically determined by

the top-$k$ entries in $T$, evolves slowly across iterations (an illustration of this phenomenon is given in Appendix A.2). To this end, we introduce an amortized symbolic analysis strategy. We perform the expensive CPU-based symbolic analysis only once every $S$ iterations (e.g., $S = 10$). For the subsequent $S - 1$ iterations, we reuse the established symbolic pattern structure, updating only the numerical values of the non-zero elements. This dramatically reduces the CPU overhead and improves the overall GPU duty cycle.

This method is simple and easy to implement, but we shall emphasize that this is not a trivial engineering modification. Clearly, reusing the sparsity pattern will change the quasi-Newton search direction in each iteration, and we must show that it does not compromise the convergence properties of the algorithm. A formal proof is presented in Section 5.2, and here we provide some insights. The validity of this modification is grounded on Theorem 4.1, which is derived from Corollary 3.4 of Wang & Qiu (2025):

**Theorem 4.1** (Rephrased Corollary 3.4 of Wang & Qiu, 2025). *For any sparsification scheme $\Omega \supseteq \Omega^*$, the sparsified Hessian matrix $H_\Omega$ satisfies*

$$0 < \lambda_{\min}(H) \le \lambda_{\min}(H_\Omega) \le \lambda_{\max}(H_\Omega) \le \lambda_{\max}(H),$$

*where $\lambda_{\min}(A)$ and $\lambda_{\max}(A)$ stand for the smallest and largest eigenvalues of a matrix $A$.*

In other words, any sparsification made to the Hessian matrix $H$ does not expand the range of its eigenvalues, so no matter we reuse the sparsity pattern or not, the condition number of $H_\Omega$ is always bounded by $\lambda_{\max}(H)/\lambda_{\min}(H)$. In theoretical analysis, the condition number of the $B$ matrix is crucial to the convergence of quasi-Newton methods. Since $B$ is strongly connected to $H_\Omega$, a control of $\lambda_{\max}(B)/\lambda_{\min}(B)$ can also be anticipated. In summary, the amortized symbolic analysis provides significant computational benefits, and meanwhile is a safe algorithmic change that possesses a solid theoretical guarantee.

### 4.3 Collaborative CPU-GPU Computing for Candidate Iterate Generation

Even with amortization, the GPU must wait for the CPU during the symbolic analysis phase every $S$-th iteration, and there is still massive computing time spent on the CPU-only part. During this period, GPU is almost idle, which potentially results in a waste of computing power. The upper part of Figure 2 visualizes this phenomenon by profiling on the compiled code of a vanilla version of the proposed solver that implements the method in Section 4.2. In most of the time, the occupacy of GPU kernels keeps staying at a high level. However, when the sparsity pattern is recomputed, there is a visible gap between kernel function calls, indicating that only CPU is working during this time.

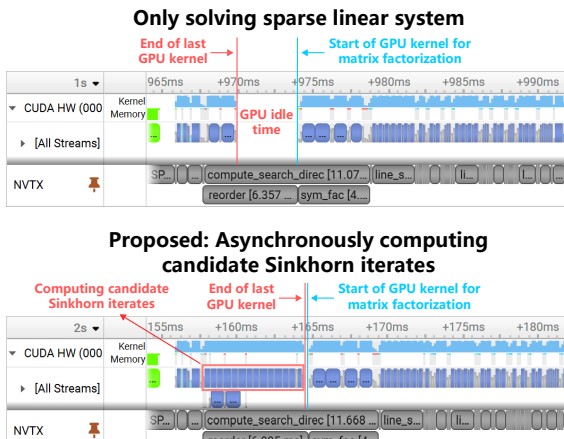

*Figure 2.* Visualization of GPU kernel occupancy.

To make the best use of this residual idle time, we propose a collaborative CPU-GPU computing mechanism. The core idea is to generate candidate iterates on GPU during the symbolic analysis part that merely utilizes CPU. Recall that the Sinkhorn algorithm solves the same dual problem (3). While generally slower to converge than Newton-type methods, its iterations are computationally inexpensive and highly parallelizable on the GPU.

Therefore, when the host CPU begins the serial symbolic analysis task, we propose to simultaneously trigger an asynchronous CUDA stream on the GPU, which performs several lightweight Sinkhorn iterations on the current dual variables, generating an alternative candidate solution $x_k^s$. Once the CPU completes the analysis and the GPU finishes the subsequent quasi-Newton step to generate the primary candidate $x_k^q$, cuRegOT evaluates the objective function and gradient norms for both candidates. It then selects the one yielding greater improvement. This strategy effectively masks the CPU latency and provides more explorations of the solution space, thus potentially resulting in faster overall convergence without incurring additional wall-clock time. The lower part of Figure 2 shows the profiling result on the improved algorithm. It can be seen that multiple Sinkhorn iterations have been computed during the symbolic analysis stage, and the runtime for this stage is almost unaffected.

We shall clarify that inserting external iteration steps into an existing optimization algorithm does not automatically guarantee the convergence of the modified procedure, especially when the insertion occurs possibly infinite times. This is because the inserted steps may disrupt the dependence structure between the original consecutive iterates, which the convergence guarantee may rely on. In the case of the proposed algorithm, however, we provide a rigorous theoretical guarantee in Section 5.2, showing that adding such candidate selection step preserves the convergence

properties of the original SPLR algorithm.

### 4.4 Fused Kernel Design for Gradient Evaluation

A naive implementation of the gradient calculation involves multiple passes over the large cost matrix $M$ and the intermediate transport plan matrix $T$ in global memory: one pass to compute $T_{ij}$ elements, and subsequent passes to reduce rows to compute $\nabla_\alpha f = T\mathbf{1}_m - a$ and columns to compute $\nabla_{\beta_{-m}} f = T_{-m}^T \mathbf{1}_n - b_{-m}$. On GPUs, this is severely bound by memory bandwidth.

We propose a fused CUDA kernel that performs element-wise computation, row reduction, and column reduction in a single pass over the data, maximizing data reuse in the fastest levels of the memory hierarchy (registers and shared memory). The kernel operates on the matrix $M$ using a grid-stride loop to handle arbitrary matrix dimensions. We deploy thread blocks of size $32 \times D$, where 32 corresponds to the CUDA warp size, and $D$ is a tunable parameter (empirically set to $D = 8$ in our implementation). Each thread block processes a $D \times 32$ tile of the matrix.

**Register-level computation**   The thread loads $\alpha_i$, $\beta_j$, and $M_{ij}$ from global memory and computes $T_{ij} = \exp\{(\alpha_i + \beta_j - M_{ij})/\eta\}$. This value is held locally in a register.

**Warp-level row reduction**   Threads within a single warp (processing the same row $i$) cooperate to sum their register values of $T_{ij}$. We utilize highly efficient warp shuffle intrinsics `__shfl_down_sync()` to perform a parallel reduction tree directly in registers, requiring zero shared memory usage. The thread at lane 0 of the warp writes the partial row sum to the global memory.

**Shared memory column reduction**   Threads within the same block column (processing the same column $j$) sum their values, utilizing the fast shared memory (L1 cache). Each thread atomically adds its $T_{ij}$ value to a shared memory buffer dedicated to its column index within the block. By choosing a small dimension $D = 8$, we ensure low bank conflict pressure during these atomic operations. Once the block finishes the grid-stride loop, it adds the partial column sums to the global memory.

This fused kernel ensures that each $M_{ij}$ is read exactly once, significantly alleviating the bandwidth bottleneck.

## 5 Theoretical Analysis

### 5.1 Computational Complexity

In this section we focus on analyzing the computational complexity of the fused kernel proposed in Section 4.4, and most of the other parts of the algorithm involve standard matrix and vector operations that have well-established complexity analyses. Suppose that there are $K$ GPU threads

that can work in parallel. In our setting, every $32D$ threads form a block of size $D \times 32$ that each time processes a tile of the $T$ matrix, and we assume that the total $K$ threads form a grid of blocks with $g_r$ rows and $g_c$ columns. In other words, a grid consists of $g_r \times g_c$ blocks, and each block contains $D \times 32$ threads. Clearly, we have $K = 32Dg_r g_c$. We summarize the computational costs for major operations in gradient evaluation in Table 1, with calculation details given in Appendix B.1. The memory-write terms are conservative estimates that account for possible serialization caused by atomic additions to the same output row/column accumulator.

*Table 1.* Computational costs for major operations in gradient evaluation.

| Operation | Hardware Level | Running Time on $K$ Processors |
|---|---|---|
| $T$ matrix elements | Register | $O(nm/K)$ |
| Row sum reduction | Register | $O(5nm/K)$ |
| | Global memory write | $O(g_c nm/K)$ |
| Column sum reduction | Register | $O(mn/K)$ |
| | Shared memory write | $O(Dm/(32g_c))$ |
| | Global memory write | $O(g_r m/(32g_c))$ |

### 5.2 Convergence of the Algorithm

As explained in Sections 4.2 and 4.3, the algorithmic improvements we have made to the SPLR algorithm do not automatically preserve the convergence properties. Therefore, we formally establish the theoretical guarantee for the modified algorithm via the following two theorems.

**Theorem 5.1.** *Let $\{x_k\}$ be a sequence of iterates such that $x_k$ is generated by the SPLR algorithm (Algorithm 1) for $k \neq iS$, $i = 1, 2, \ldots$, where $S \geq 2$ is a pre-specified integer. For $k = iS$, $i = 1, 2, \ldots$, let $x_k^q$ be the next iterate of $x_{k-1}$ using Algorithm 1, and $x_k^s$ be an arbitrary point. We set $x_k = x_k^s$ if $f(x_k^s) \leq f(x_k^q)$, and take $x_k = x_k^q$ otherwise. Then we have*

$$\lim_{k \to \infty} \|\nabla f(x_k)\| = 0.$$

In other words, we have shown that the sequence $\{x_k\}$ is globally convergent to an optimal point of $f(x)$. Moreover, the modified algorithm still has a linear convergence rate.

**Theorem 5.2.** *Under the same setting as Theorem 5.1, the iterates $\{x_k\}$ at least have a linear convergence rate. That is, there exists a constant $0 < r < 1$ such that*

$$f(x_k) - f^* \leq r\left[f(x_{k-1}) - f^*\right]$$

*for all $k \geq 1$, where $f^*$ is the optimal objective function value of $f(x)$.*

The expression of $r$ is given in Appendix B.3, and we note that it depends on the problem parameters $(M, a, b, \eta, x_0)$.

Equipped with the two theorems, the algorithmic designs in Section 4 not only provide practical engineering optimizations, but also enjoy rigorous theoretical guarantees.

# 6 Numerical Experiments

## 6.1 Overview

In this section, we evaluate the performance of different GPU-based OT solvers on a variety of benchmark problems. For each problem instance, we generate the cost matrix $M \in \mathbb{R}^{n \times m}$, the source distribution vector $a \in \mathbb{R}_+^n$, and the target distribution vector $b \in \mathbb{R}_+^m$, where $\sum_{i=1}^n a_i = \sum_{j=1}^m b_j = 1$. Following standard practice in entropic-regularized OT literature, we normalize the cost matrix by its maximum value, *i.e.*, $M \leftarrow M / \max_{i,j} M_{ij}$, to make the regularization parameter $\eta$ comparable across test examples. We fix $\eta = 0.001$ in the main experiments, and explore the impact of different $\eta$ values in Appendix A.6.

The GPU solvers included in our experiments are the POT package (Flamary et al., 2021) with two algorithms, Sinkhorn and Greenkhorn, the OTT-JAX package (Cuturi et al., 2022) with the Sinkhorn algorithm and its Anderson acceleration version, a CuPy implementation of the accelerated Sinkhorn algorithm (AccSinkhorn) proposed in Lin et al. (2022), and the proposed cuRegOT solver. The memory size for Anderson acceleration is set to 5, and we fix the hyperparameter $S$ in cuRegOT to $S = 10$ for the experiments. In Appendix A.7, we additionally explore the impact of different $S$ values in the cuRegOT solver.

For each solver, we set the convergence tolerance to zero and run for some fixed numbers of iterations (typically 10, 20, 50, 100, etc.) to obtain detailed convergence trajectories. This approach allows us to compare the efficiency of different solvers without being confounded by their different stopping criteria. For each test problem and solver configuration, we first run the solver for a few iterations to warm up before measuring their actual runtime, and we repeat the experiment ten times to account for runtime variability. We report the median runtime and median optimization error across these runs, where the optimization error is measured by the marginal error of the computed transport plan $T$: Error $= \|T\mathbf{1}_m - a\|_1 + \|T^T\mathbf{1}_n - b\|_1$. The rationale of using the marginal error to quantify optimality in our setting is explained in Appendix A.5, where we also consider the duality gap as an alternative metric. Finally, we visualize the results by plotting the optimization error (on a logarithmic scale) against runtime for each solver, which reveals both

the convergence speed and final accuracy of different algorithms. Lower curves indicate better performance, achieving lower errors in less time.

## 6.2 Synthetic Data

We construct three types of synthetic benchmark problems to evaluate the solver performance across different geometric structures and distribution properties. For each type, multiple problem sizes of $(n, m) = (1600, 1200), (3200, 2400), (6400, 4800)$ are tested to assess scalability.

**Synthetic I: Gaussian point clouds** We generate two samples of Gaussian distributions to create cost matrices based on squared Euclidean distances. We consider two variants:

- *Same distribution (iid)*: entries of both source and target samples are independently drawn from $N(0, 1)$, thus testing solvers on symmetric, well-conditioned problems.
- *Different distributions (diff)*: source and target samples have $N(0, 1)$ and $N(1, 0.25)$ entries, respectively. This setting creates asymmetric problems that require more substantial mass transport.

For both variants, the source and target distribution vectors $a$ and $b$ are uniform.

**Synthetic II: Exponential to mixture Gaussian transport** This benchmark models transport between (discretized) continuous probability distributions. The source distribution is an exponential distribution with mean one, admitting the density function $a(x) = e^{-x}$, while the target distribution is a mixture of two Gaussians: $b(x) = 0.2 \cdot N(x; 1, 0.04) + 0.8 \cdot N(x; 3, 0.25)$. We discretize both distributions on a uniform grid of points $x_1, \ldots, x_n$ and $y_1, \ldots, y_m$ on $[0, 5]$, respectively, and compute the squared Euclidean distance $|x_i - y_j|^2$ as the $(i, j)$ entry of the cost matrix. This problem tests solvers on problems where the OT transport plan exhibits a multi-modal structure.

## 6.3 CIFAR-10 Data

To evaluate solvers on real-world computer vision tasks, we construct OT problems using the CIFAR-10 image dataset (Krizhevsky, 2009). CIFAR-10 contains 60000 color images of size $32 \times 32 \times 3$ across 10 classes, with 5000 training images per class. For each benchmark instance, we select two classes from CIFAR-10 as the source and target distributions, where each image is a discrete point in the distribution. This creates problems with $n = m = 5000$. For the cost matrix, we do not directly use raw pixel values, but instead extract deep features using a pretrained ResNet-18 model (He et al., 2016). The preprocessing details are described in Appendix A.3. For each pair of source and target images, we compute the squared Euclidean distance between their fea-

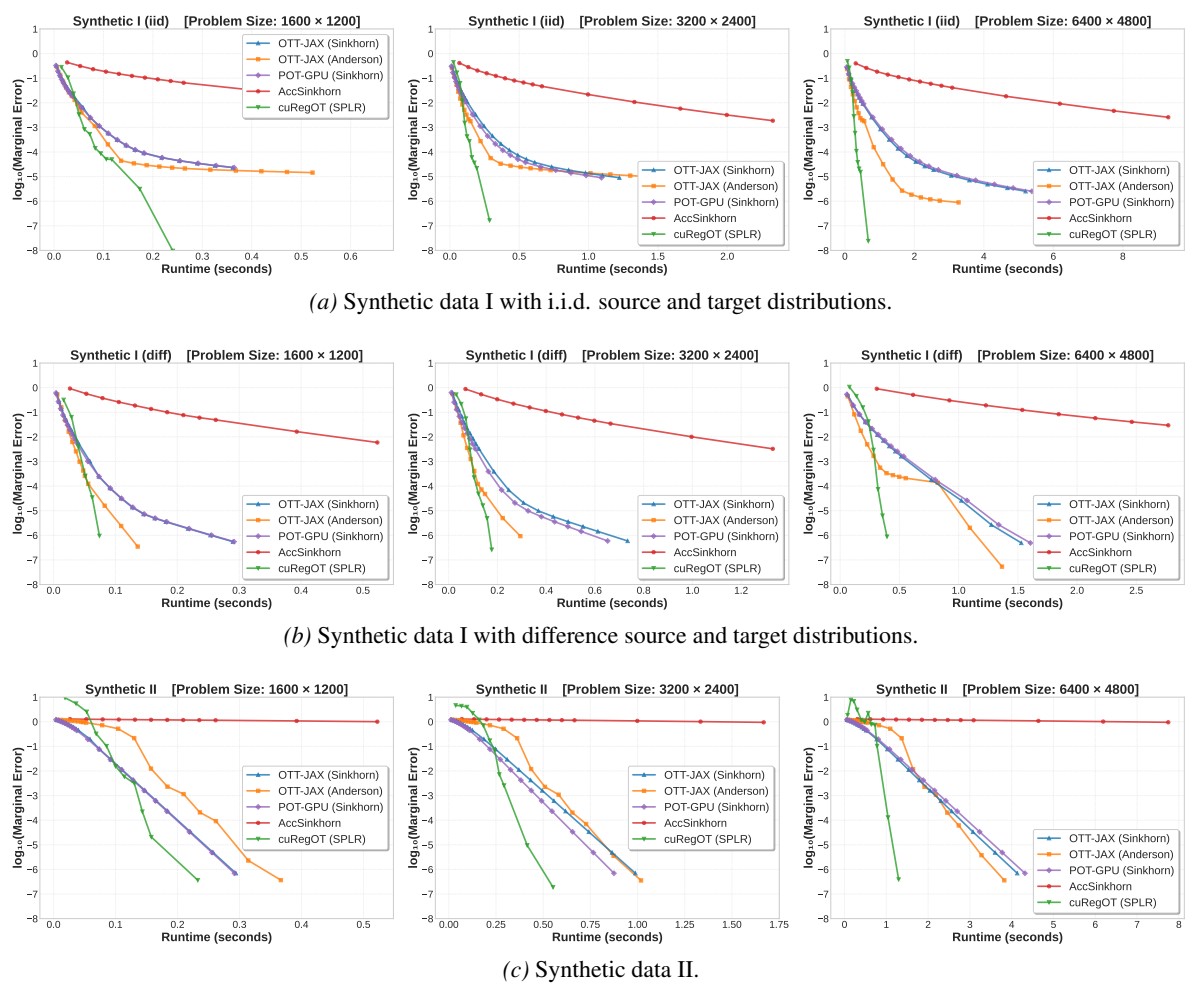

*(a)* Synthetic data I with i.i.d. source and target distributions.

*(b)* Synthetic data I with difference source and target distributions.

*(c)* Synthetic data II.

*Figure 3.* Benchmark result for synthetic datasets. The horizontal axis represents the elapsed wall time, and the vertical axis is the optimization error on a logarithmic scale.

ture vectors to construct the cost matrix $M \in \mathbb{R}^{5000 \times 5000}$, and both marginal distributions $a$ and $b$ are set to uniform.

This benchmark tests solvers on large-scale, real-world problems where the cost structure reflects high-level semantic relationships rather than simple geometric distances.

### 6.4 Result

In all experiments, we have found that the Greenkhorn algorithm has a quite slow convergence speed, exhibiting almost flat curves, so we exclude it in all the plots. The experiment results for the synthetic data and the CIFAR-10 data are visualized in Figures 3 and 4, respectively. First of all, we can observe that the two implementations of the Sinkhorn algorithm are close to each other, suggesting that their performance are representative. For the Anderson acceleration, it provides visible speedup to the standard Sinkhorn algorithm in the Synthetic I and CIFAR-10 cases, but still generally faces a slow convergence to a high precision. On the other

hand, it can be even slower in the Synthetic II case.

For the proposed cuRegOT solver, it is clear that it exhibits a significantly faster convergence in almost all cases, especially when a relatively high precision is requested. Moreover, the advantage of cuRegOT tends to be larger for bigger problem sizes, further validating its scalability for large-scale and real-word problems.

### 6.5 Ablation Study

To study the effectiveness and necessity of the algorithm improvements introduced in Section 4, we conduct an ablation study by testing the solver performance after removing one or more components of the algorithm design. The results are shown in Figure 5, in which "-A" means removing the amortized symbolic analysis strategy, "-S" means removing the candidate Sinkhorn iterate generation, and "-A-S" means removing both. Clearly, the plots show that removing either of the algorithmic improvement may slow down

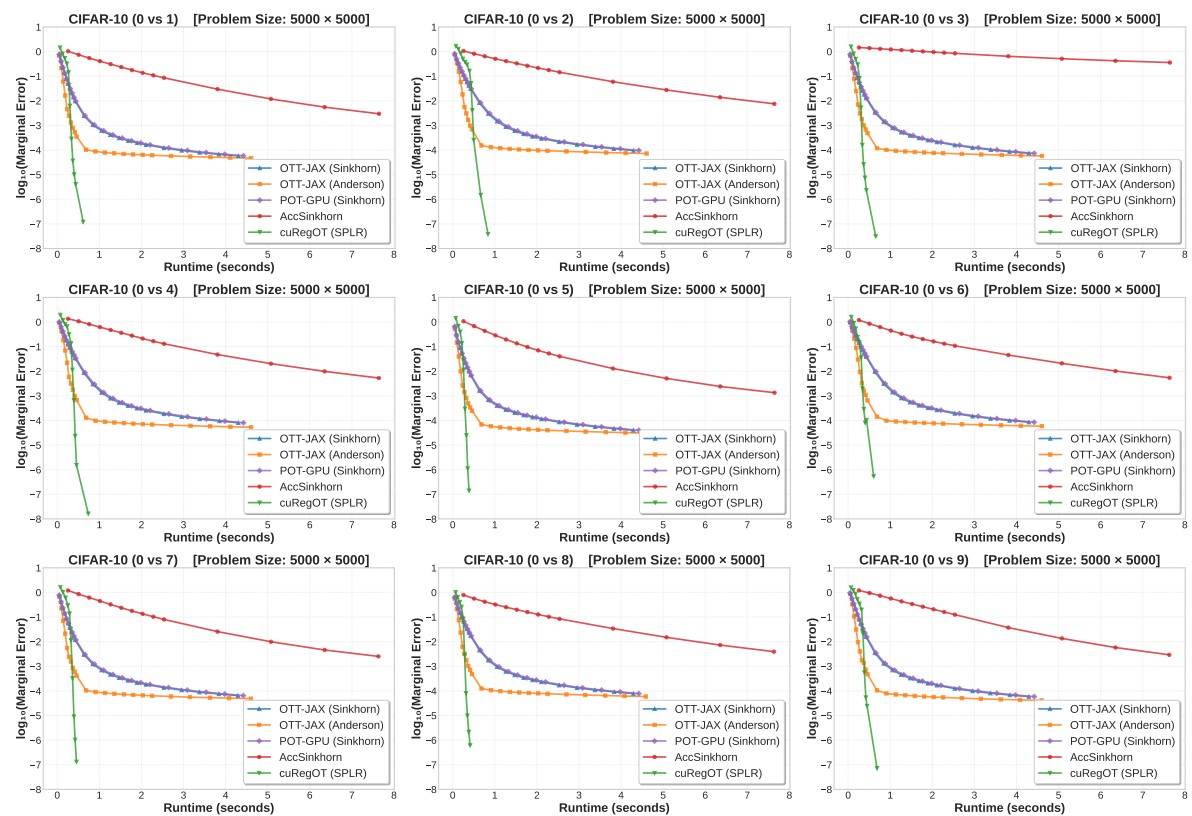

*Figure 4.* Benchmark result for the CIFAR-10 dataset.

the solver. This suggests that both the amortized symbolic analysis and the Sinkhorn iteration generation contribute to the performance of the cuRegOT solver.

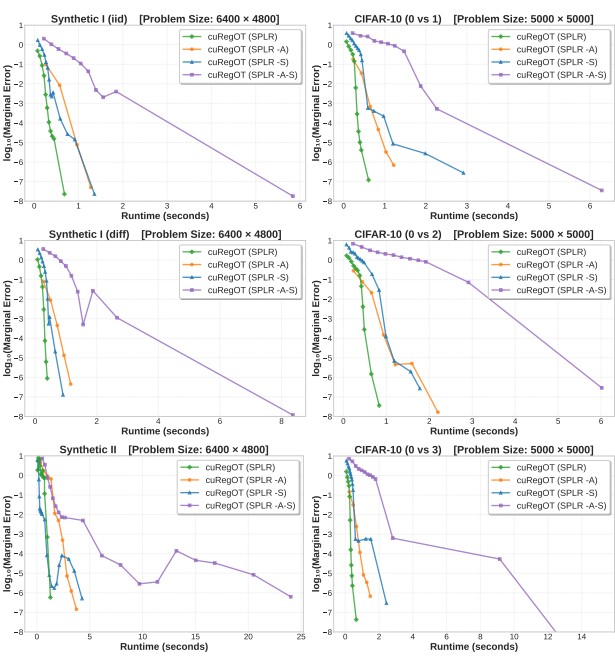

*Figure 5.* Ablation study on the algorithm design.

# 7 Conclusion

This paper presented cuRegOT, a high-performance GPU solver for entropic-regularized OT, built upon the SPLR quasi-Newton method. The main obstacle in bringing Newton-type methods to GPUs are sparse linear systems: the symbolic analysis and reordering needed by the sparse Cholesky decomposition are difficult to parallelize and are often executed on the CPU. To address this bottleneck, cuRegOT introduces three GPU-oriented designs: (1) *amortized symbolic analysis* by reusing sparsity patterns across multiple iterations; (2) *collaborative CPU–GPU computing* that overlaps host-side analysis with GPU Sinkhorn-based candidate generation; and (3) *fused CUDA kernel* for efficient gradient evaluation to reduce memory traffic.

We established rigorous guarantees showing that these modifications preserve the global convergence of the original SPLR method and retain at least a linear convergence rate. In numerical experiments, cuRegOT consistently achieves faster convergence than GPU Sinkhorn baselines (including Anderson acceleration), with larger gains at higher accuracy targets and larger problem sizes. Future work includes further reducing reliance on host-side symbolic steps, exploring iterative or mixed-precision sparse solvers, and extending the approach to broader OT formulations.

## Acknowledgements

Yixuan Qiu's work was supported in part by National Natural Science Foundation of China (72571163), Shanghai Pujiang Program (21PJC056), and Shanghai Engineering Research Center of Finance Intelligence (19DZ2254600).

## Impact Statement

This paper presents work whose goal is to advance the field of machine learning. There are many potential societal consequences of our work, none of which we feel must be specifically highlighted here.

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

# A   Additional Experiment Details

## A.1   Computing Environment

The numerical experiments are conducted on a workstation with an AMD 5900X CPU and an Nvidia RTX 6000 Ada GPU. The workstation runs on a Ubuntu 25.10 Linux operating system and the CUDA 13.2 platform.

## A.2   Sparsity Pattern of the Transport Plan Across Iterations

To support the claim in Section 4.2 that the sparsity pattern of the $T(x)$ matrix changes slowly across iterations, below we use a motivating example to visualize the evolution of the transport plan $T(x)$ in a realization of the Sinkhorn algorithm. The OT task is defined by the Synthetic II problem introduced in Section 6.2, with the problem size $n = m = 32$ and a regularization parameter $\eta = 0.001$. We compute the $T(x)$ matrix every ten iterations, and visualize it using a heatmap, as illustrated in Figure 6. It can be observed that the top-$k$ entries of $T(x)$ show smooth changes across iterations, suggesting that we can reuse the sparsity pattern to reduce symbolic analysis overhead.

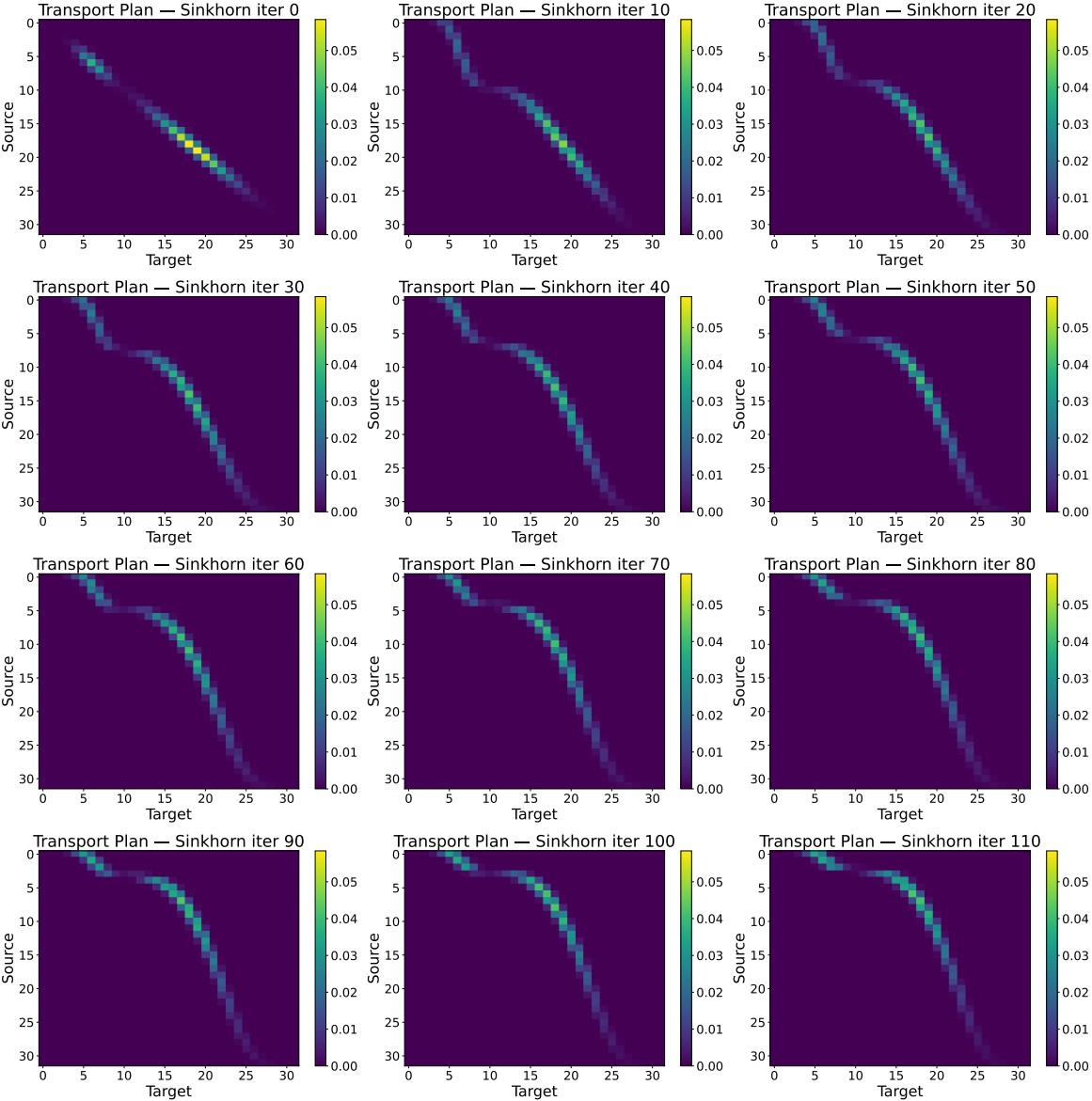

*Figure 6.* Evolution of the transport plan during Sinkhorn iterations, based on the Synthetic II problem introduced in Section 6.2 with $n = m = 32$ and $\eta = 0.001$.

## A.3 Preprocessing of the CIFAR-10 Data

**Problem construction** For each benchmark instance, we select two classes from CIFAR-10 as the source and target distributions. For example, we transport between class 0 (airplane) and class 1 (automobile), or class 3 (cat) and class 5 (dog). This creates problems with $n = m = 5000$, where each point represents an image.

**Feature extraction** Directly using raw pixel values ($32 \times 32 \times 3 = 3072$ dimensions) for distance computation is inadequate, as pixel-wise Euclidean distance does not capture semantic similarity between images. Instead, we extract deep features using a pretrained ResNet-18 model (He et al., 2016) trained on the ImageNet dataset (Deng et al., 2009). Specifically:

1. Each CIFAR-10 image is upsampled from $32 \times 32 \times 3$ to $224 \times 224 \times 3$ to match the ImageNet input size expected by ResNet-18.
2. Images are normalized using ImageNet's mean and standard deviation.
3. We remove the final classification layer of ResNet-18 and extract the 512-dimensional feature vectors from the penultimate layer (after global average pooling).

This feature extraction pipeline produces semantically meaningful embeddings where distances reflect perceptual similarity between images.

**Cost matrix construction** For each pair of source and target images, we compute the squared Euclidean distance between their feature vectors to construct the cost matrix $M \in \mathbb{R}^{5000 \times 5000}$. Both marginal distributions $a$ and $b$ are set to uniform.

## A.4 Additional Test Cases

Figure 7 shows the benchmark result on more test cases using the CIFAR-10 dataset.

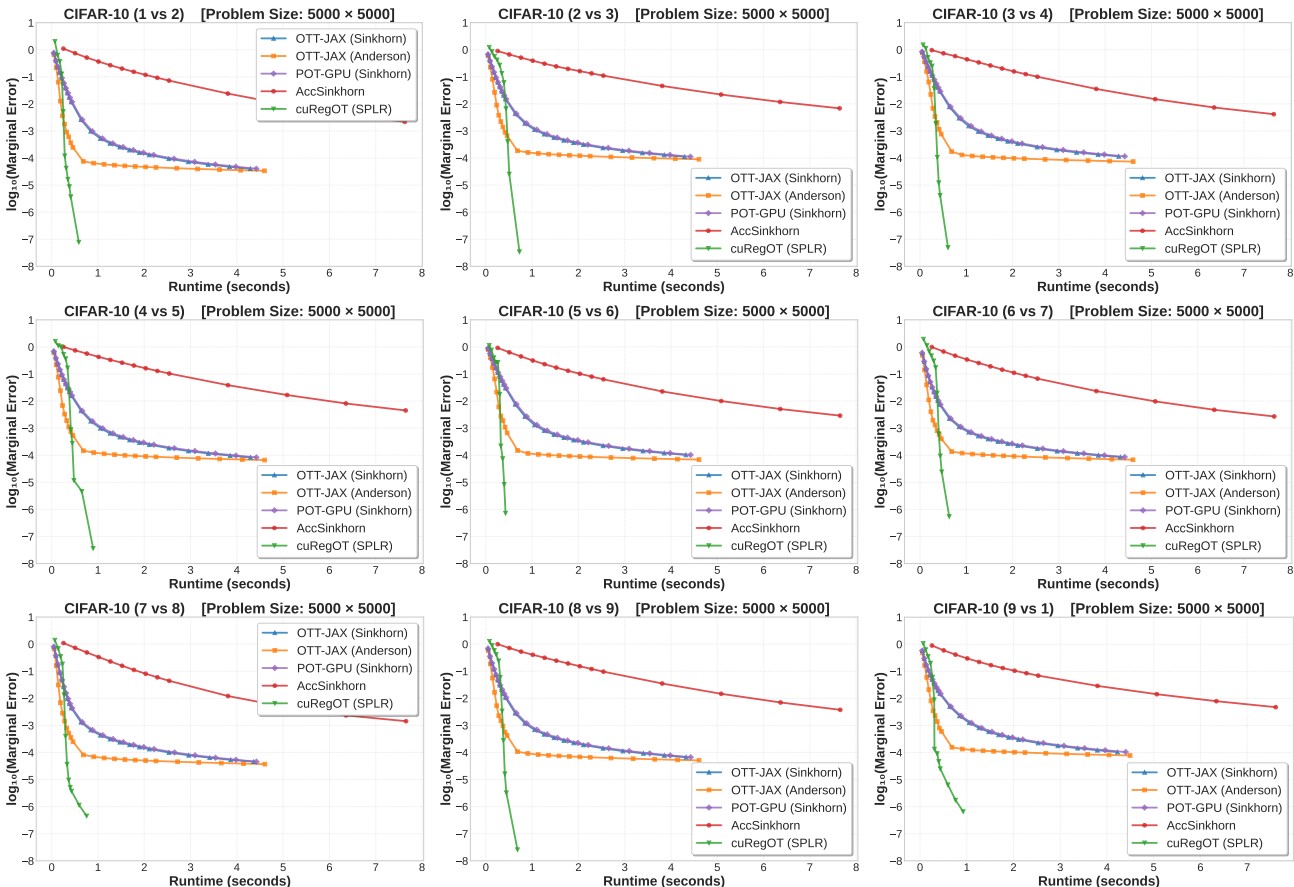

*Figure 7.* Benchmark result for the CIFAR-10 dataset with additional class pairs.

### A.5 Using Duality Gap as the Optimization Error Metric

In the numerical experiments, we have used the marginal error to quantify the optimization error. For general algorithms, only testing the marginal error is not sufficient, as the trivial transport plan output $ab^T$ results in a zero marginal error. However, the methods compared in this paper are all based on the dual problem (3), meaning that the output transport plan is obtained as $T_{ij} = \exp((\alpha_i + \beta_j - M_{ij})/\eta)$. Under this special structure, the marginal error is exactly the gradient norm of the dual problem. Therefore, testing the gradient is a valid indicator of the optimality.

In addition to the marginal error, we also consider the duality gap as a metric to evaluate the performance of different solvers. By definition, the duality gap is the difference between the current primal value $L_p$ and the current dual value $L_d$, where

$$L_p = \langle T, M \rangle - \eta \cdot h(T) = \langle T, M \rangle + \eta \cdot \sum_{i,j} T_{ij} \cdot \log(T_{ij}) - \eta \cdot \sum_{i,j} T_{ij},$$

$$L_d = -\eta \cdot \sum_{ij} T_{ij} + \alpha^T a + \beta^T b,$$

and $T_{ij} = \exp\{\eta^{-1}(\alpha_i + \beta_j - M_{ij})\}$. After some simplification, we can obtain that

$$L_p - L_d = \langle T, M \rangle + \sum_{i,j} T_{ij}(\alpha_i + \beta_j - M_{ij}) - \alpha^T a - \beta^T b$$

$$= \sum_{i,j} T_{ij}(\alpha_i + \beta_j) - \alpha^T a - \beta^T b$$

$$= \alpha^T (T\mathbf{1}_m - a) + \beta^T (T^T \mathbf{1}_n - b).$$

In Figure 8, we use $\log_{10}|L_p - L_d|$ as the optimality metric, and plot it against the wall time. The plots show that the duality gap demonstrates very similar patterns as the marginal error. In fact,

$$|L_p - L_d| \leq \|\alpha\|_\infty \|T\mathbf{1}_m - a\|_1 + \|\beta\|_\infty \|T^T\mathbf{1}_n - b\|_1$$

$$\leq \max\{\|\alpha\|_\infty, \|\beta\|_\infty\} \cdot (\|T\mathbf{1}_m - a\|_1 + \|T^T\mathbf{1}_n - b\|_1),$$

where $\|T\mathbf{1}_m - a\|_1 + \|T^T\mathbf{1}_n - b\|_1$ is exactly the marginal error. Therefore, in some sense, the marginal error provides an upper bound for the duality gap, if we assume that $\|\alpha\|_\infty$ and $\|\beta\|_\infty$ can be properly bounded.

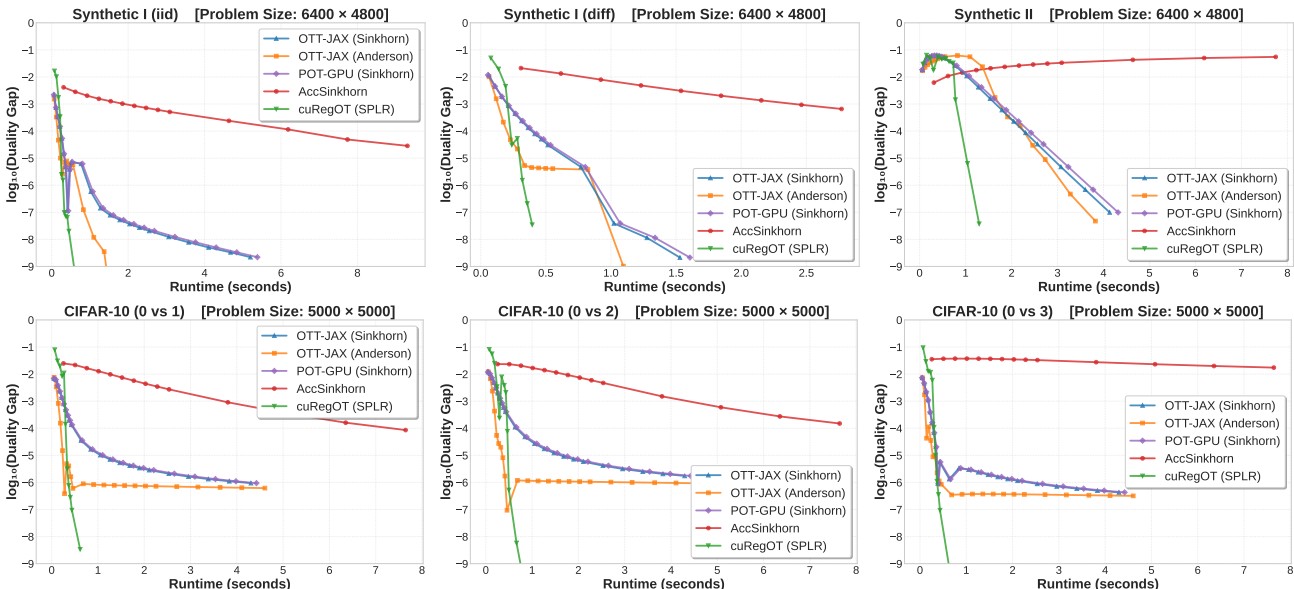

**Figure 8.** Benchmark result that uses the duality gap (on a logarithmic scale) as the metric for optimization error. The regularization parameter is set to $\eta = 0.001$.

## A.6 Impact of the Regularization Parameter $\eta$

For the experiments in the main article, we have fixed the regularization parameter to $\eta = 0.001$. To test solver's performance under different $\eta$ values, we additionally conduct experiments evaluating $\eta = 0.01$ and $\eta = 0.0001$, with results given in Figures 9 and 10, respectively. It can be observed that as $\eta$ decreases, all methods slow down due to ill-conditioning, but cuRegOT's advantage becomes more pronounced: the plots show that while Sinkhorn and its variants deteriorate rapidly at smaller $\eta$ values, cuRegOT maintains a fast convergence speed. This indicates that the second-order curvature information we utilize is practically indispensable for overcoming the ill-conditioning caused by small $\eta$.

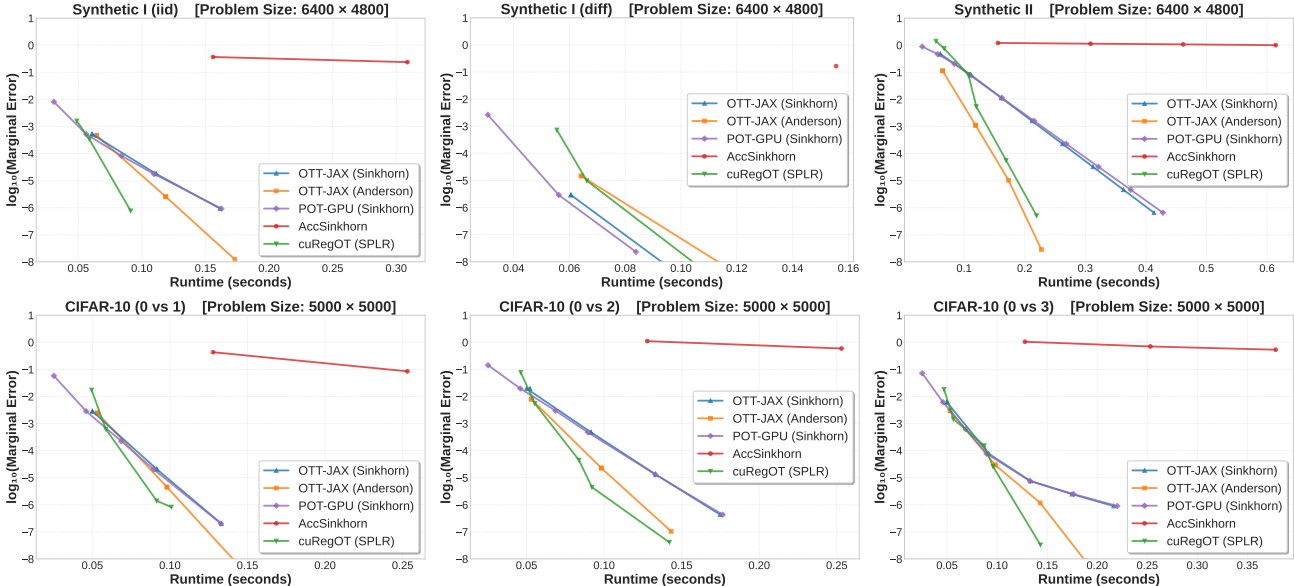

*Figure 9.* Benchmark result for different solvers under the regularization parameter $\eta = 0.01$.

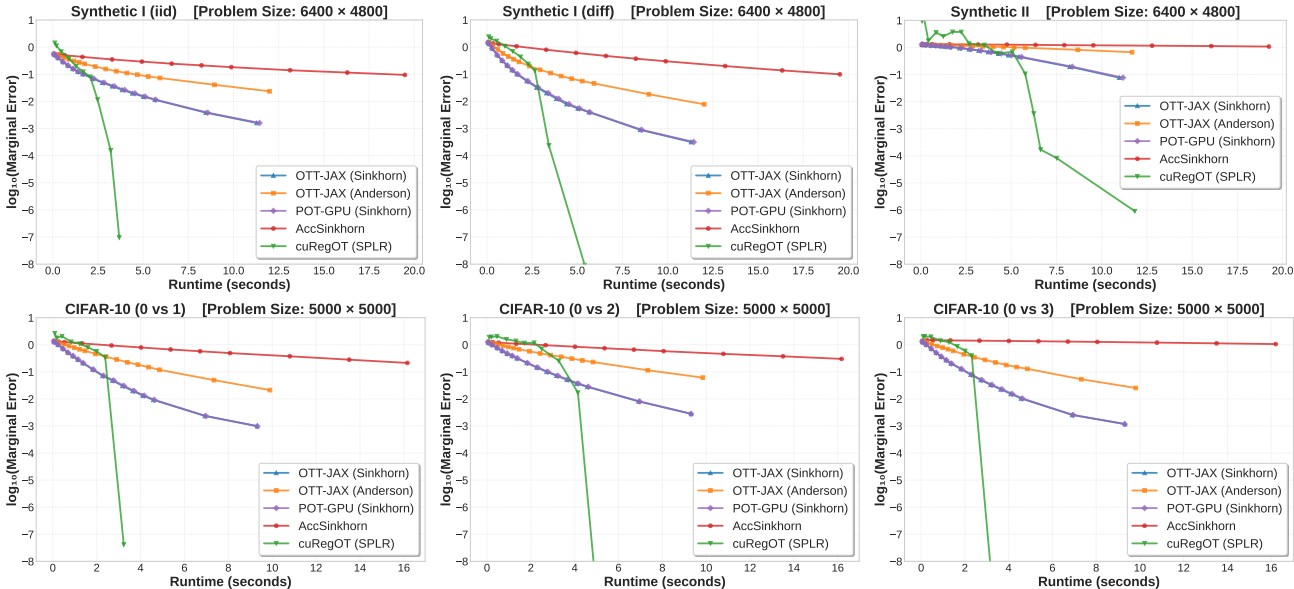

*Figure 10.* Benchmark result for different solvers under the regularization parameter $\eta = 0.0001$.

### A.7    Impact of the Hyperparameter $S$

The main hyperparameter in the cuRegOT solver is $S$, the number of iterations for which we reuse the sparsity pattern (and thus reuse symbolic analysis). In Figure 11 we conduct an $S$-sensitivity study to examine the impact of $S$ on the solver performance. We find that the method is relatively robust: $S \in [5, 30]$ consistently yields good accuracy-time curves across problems, while $S = 1$ (no reuse) can be slower due to CPU overhead.

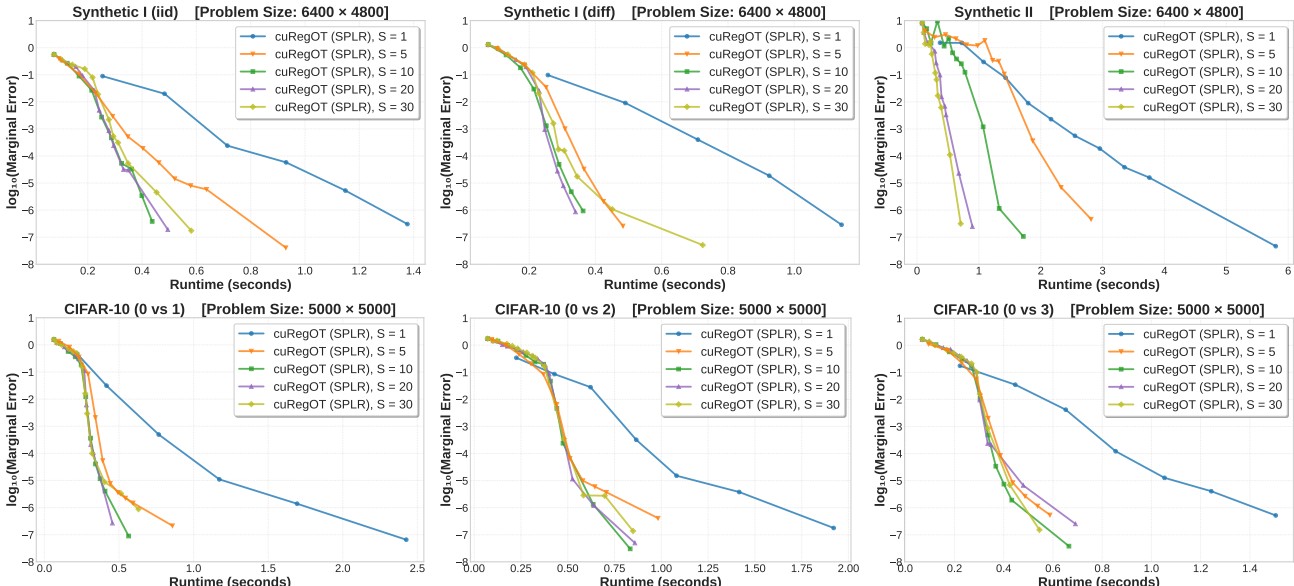

*Figure 11.* Benchmark result for the hyperparameter $S$ in the cuRegOT solver. The regularization parameter is set to $\eta = 0.001$.

## B    Theoretical Analysis Details

### B.1    Complexity Analysis for Table 1

Each row of the $D \times 32$ thread block forms a warp, and it can compute the block row sum using 5 parallel additions. Moreover, each thread block needs to process $nm/K$ tiles of the matrix, so computing the block-level row sums for the whole $T$ matrix costs $O(5nm/K)$ wall time.

Each warp produces one partial row sum for a 32-column segment and atomically accumulates it into the corresponding row sum entry in the global memory. For each row, there are roughly $m/32$ such partial sums. Although different column blocks are launched in parallel, their atomic additions target the same row sum address, and we use a conservative model in which these updates are serialized. Since $g_r D$ rows can be processed simultaneously, the global memory writing time for row sums is

$$O\left(\frac{n}{g_r D} \cdot \frac{m}{32}\right) = O\left(\frac{g_c nm}{K}\right).$$

Computing the column sums consists of two stages. In the first stage, each block adds the $T_{ij}$ elements in the same column to the shared memory, and in the second stage, the block partial column sums are added to the global memory. In the first stage, each thread sequentially processes $n/(g_r D)$ rows of $T$, leading to $O(n/(g_r D))$ register-level additions. Then within the block, adding thread-owned values to the shared memory costs $O(D)$ time. Next, for each column, $g_r$ block-owned partial column sums are atomically accumulated into the same column sum entry in the global memory, yielding the conservative estimate of $O(g_r)$ time. Since the $K$ threads can process $32g_c$ columns simultaneously, in total, computing the column sums requires $O(m/(32g_c) \cdot n/(g_r D)) = O(mn/K)$ additions, $O(Dm/(32g_c))$ shared memory writes, and $O(g_r m/(32g_c))$ global memory writes.

### B.2   Proof of Theorem 5.1

Let $x_0$ be the initial value of the iterates $\{x_k\}$, and define the level set $\mathsf{L} = \{x : f(x) \leq f(x_0)\}$. Clearly, $\mathsf{L}$ is a convex and compact set. Since $f(x)$ is twice differentiable on $\mathsf{L}$ and $H(x) := \nabla^2 f(x)$ is positive definite, we have that the eigenvalues of $H(x)$ must be bounded on $\mathsf{L}$. That is, there exists constants $0 < \mu \leq L < \infty$ such that $\mu \leq \lambda_{\min}(H(x)) \leq \lambda_{\max}(H(x)) \leq L$ for all $x \in \mathsf{L}$, where $\lambda_{\min}(A)$ and $\lambda_{\max}(A)$ denote the smallest and largest eigenvalues of a symmetric matrix $A$, respectively. This implies that $\nabla f$ is $L$-Lipschitz, *i.e.*,

$$\|\nabla f(x) - \nabla f(y)\| \leq L\|x - y\|, \quad \forall x, y \in \mathsf{L}, \tag{9}$$

and that the Polyak-Łojasiewicz condition holds:

$$\|\nabla f(x)\|^2 \geq 2\mu[f(x) - f^*], \quad \forall x \in \mathsf{L}, \tag{10}$$

where $f^*$ is the optimal value of $f(x)$.

Let $x^-, x \in \mathsf{L}$ be two arbitrary points in the level set $\mathsf{L}$, and let $x^+$ be the output of Algorithm 1 applied to $x$. Clearly, $x^+ = x - \gamma B^{-1} g$, where $g = \nabla f(x)$ and $B = H_\Omega + \xi u u^T + \zeta v v^T + \tau I$ as defined in (5) and (6). We show next that the eigenvalues of $B$ are also bounded on $\mathsf{L}$, uniformly on the values of $x^-$ and $x$.

To start with, Corollary 3.4 of Wang & Qiu (2025) shows that $\lambda_{\min}(H) \leq \lambda_{\min}(H_\Omega) \leq \lambda_{\max}(H_\Omega) \leq \lambda_{\max}(H)$, $H = H(x)$, so we have $\mu \leq \lambda_{\max}(H_\Omega) \leq L$. In case $(y^-)^T s^- \leq 0$, Algorithm 1 forces $B = H_\Omega + \tau I$, so we easily obtain

$$\mu \leq \lambda_{\min}(H_\Omega) \leq \lambda_{\min}(B) \leq \lambda_{\max}(H_\Omega) + \tau \leq L + \tau_{\max}.$$

For $(y^-)^T s^- > 0$, the proof of Theorem 5.1 of Wang & Qiu (2025) applies here, so by that theorem, we have

$$m_B := (2 + 3L/\mu)^{-1}\mu \leq \lambda_{\min}(B) \leq \lambda_{\max}(B) \leq M_B := 2L + \tau_{\max}.$$

Clearly, the bounds $(m_B, M_B)$ also hold for the $(y^-)^T s^- \leq 0$ case, and they do not depend on the values of $x^-$ and $x$.

Next, we prove that there exists a constant $C > 0$ such that

$$f(x^+) \leq f(x) - C\|g\|^2, \tag{11}$$

and $C$ does not depend on the values of $x^-$ and $x$.

To see this, define $\phi(\gamma) = f(x + \gamma d)$, where $d = -B^{-1} g$. Since $\nabla f$ is $L$-Lipschitz by (9), we have

$$|\phi'(\gamma) - \phi'(0)| = |[\nabla f(x + \gamma d) - \nabla f(x)]^T d| \leq \|\nabla f(x + \gamma d) - \nabla f(x)\| \cdot \|d\| \leq L\gamma\|d\|^2$$

using the Cauchy–Schwarz inequality. The Wolfe conditions (8) guarantees that

$$\phi'(\gamma) - \phi'(0) = (g^+ - g)^T d \geq (c_2 - 1)g^T d = (1 - c_2)|\phi'(0)|,$$

where $g^+ = \nabla f(x^+)$. Then by combining the two inequalities above, we have

$$\gamma \geq \frac{(1 - c_2)|\phi'(0)|}{L\|d\|^2}.$$

Also by the Wolfe conditions, we have

$$f(x^+) \leq f(x) + c_1 \gamma g^T d = f(x) - c_1 \gamma |\phi'(0)|,$$

so we get

$$f(x^+) \leq f(x) - \frac{c_1(1 - c_2)|\phi'(0)|^2}{L\|d\|^2}.$$

Notice that $|\phi'(0)| = -g^T d = g^T B^{-1} g \geq M_B^{-1}\|g\|^2$ and $\|d\| = \|B^{-1} g\| \leq m_B^{-1}\|g\|$, so we then obtain

$$\frac{|\phi'(0)|}{\|d\|} \geq \frac{M_B^{-1}\|g\|^2}{m_B^{-1}\|g\|} = \frac{m_B}{M_B}\|g\|,$$

which immediately gives (11) with

$$C = \frac{c_1(1 - c_2)m_B^2}{LM_B^2}.$$

Then we are ready to prove the main theorem. For $k = iS$, $i = 1, 2, \ldots$, we first note that $f(x_k^q) \leq f(x_{k-1})$, since $x_k^q$ is generated by the line search procedure that guarantees the decrease on objective function value. Then by the design of the algorithm, we can show that $f(x_k)$ is non-increasing on $k$. This implies that all the iterates $\{x_k\}$ stay in the level set L. Then by (11), we have that for all $k \geq 1$,

$$f(x_k) \leq f(x_k^q) \leq f(x_{k-1}) - C\|\nabla f(x_{k-1})\|^2. \tag{12}$$

Summing (12) over $k$, and then we have

$$\sum_{k=0}^{\infty} \|\nabla f(x_k)\| \leq \frac{f(x_0) - f^*}{C} < \infty,$$

which gives the desired conclusion that $\|\nabla f(x_k)\| \to 0$ as $k \to \infty$.

### B.3 Proof of Theorem 5.2

Using the same definitions for $x^-$, $x$, and $x^+$ as in the proof of Theorem 5.1, we will prove that there exists a constant $0 < r < 1$ such that

$$f(x^+) - f^* \leq r[f(x) - f^*], \tag{13}$$

and $r$ does not depend on the values of $x^-$ and $x$.

Indeed, in the proof of Theorem 5.1 we prove that

$$f(x^+) - f^* \leq f(x) - f^* - C\|\nabla f(x)\|^2.$$

By the Polyak-Łojasiewicz condition (10), we have

$$\|\nabla f(x)\|^2 \geq 2\mu[f(x) - f^*].$$

Therefore, it is easy to get

$$f(x^+) - f^* \leq f(x) - f^* - 2C\mu[f(x) - f^*] = (1 - 2C\mu)[f(x) - f^*],$$

which implies (13) with $r = 1 - 2C\mu$.

For $k = iS$, $i = 1, 2 \ldots$, we already have

$$f(x_k^q) - f^* \leq r[f(x_{k-1}) - f^*].$$

Since by construction we have $f(x_k) \leq f(x_k^q)$, we immediately get

$$f(x_k) - f^* \leq r[f(x_{k-1}) - f^*]. \tag{14}$$

The iterates $\{x_k\}$ for $k \neq iS$, $i = 1, 2, \ldots$ are generated by Algorithm 1, so they also satisfy (14). Overall, the inequality (14) holds for all iterates, and hence the desired result holds.

