# OpenReview forum: "cuRegOT: A GPU-Accelerated Solver for Entropic-Regularized Optimal Transport"
_ICML.cc/2026/Conference — ICML 2026 regular_

### Official Review · Reviewer_PDeW · 2026-03-07

**Soundness:** 2
**Presentation:** 3
**Significance:** 2
**Originality:** 2
**Overall Recommendation:** 3
**Confidence:** 3

**Summary:**

The paper studies how to make a known second-order solver for entropic discrete optimal transport run efficiently on GPUs. The main contribution is a set of implementation ideas aimed at reducing sparse linear algebra overhead and better using CPU/GPU parallelism. The authors also provide convergence claims for their modified solver and show empirical speedups over Sinkhorn-based GPU baselines on synthetic and image-based transport tasks.

**Compliance With Llm Reviewing Policy:**

Affirmed.

**Final Justification:**

In light of  the discussion with the authors during the rebuttal period aclarifications, I raised my score from 2: Reject to 3: weak reject, since the dependence in $\eta^{-1}$ are now known. While they are expected, not knowing them could be misleading to the readers.
During the discussion, I also decreased my confidence to 3 rather than 4, since I might be more on the theoretical side of algorithmic OT, where one wants improved complexities for new algorithms, rather than pure algorithmic design improvement.

I though not raise my score beyond that, since I believe my two main concerns remain valid:

- If this method is to be viewed as a viable replacement for Sinkhorn in GPU-based solvers, I believe it should offer stronger theoretical complexity guarantees than Sinkhorn. This is particularly relevant  given that several more recent solvers already provide such guarantees, although to the best of my knowledge, many of them still lack efficient GPU implementations.

- To make a convincing case that this algorithm should be chosen as *the* replacement for Sinkhorn, the paper would benefit from a broader comparison with other methods that have been proposed as superior alternatives to Sinkhorn.

**Key Questions For Authors:**

1. How does the claimed linear convergence rate depend on the entropic regularization parameter$\varepsilon$? If the constants are very unfavorable in $\varepsilon$, the practical meaning of the theorem is limited.

**Limitations:**

The paper should more clearly acknowledge that the contribution is mainly implementation-focused and that the total theoretical complexity of the scheme is unknown.

**Strengths And Weaknesses:**

The paper addresses a relevant practical problem: making high-accuracy regularized OT solvers usable on GPU hardware. The implementation ideas are reasonable and the empirical results suggest the method can outperform standard Sinkhorn baselines in the tested settings.

My main concern is that the paper does not clearly characterize the overall complexity of the algorithm.
More precisely, I am not convinced by the theoretical contribution in its current form. The paper gives a linear convergence statement of the form “there exists $r>0$ such that the contraction is $1-r$,” but in discrete entropic OT such rates are often known to have very poor  dependence on the regularization parameter $\varepsilon$ (exponentially bad). Without making this dependence explicit, the total theoretical computational complexity is unknown.

The experimental section is too narrow for the claim being made. The method is compared mainly against Sinkhorn variants, while there are now several accelerated OT methods with better convergence guarantees and stronger practical behavior. As a result, the paper shows that this GPU implementation can beat Sinkhorn baselines, but it does not establish competitiveness against stronger alternatives.

Finally, I find the originality limited. The core solver is not new; the main contribution is a GPU implementation and engineering optimization of a known method. That may be a solid systems contribution, but in my view it is not enough for ICML without stronger conceptual novelty, broader comparisons, or a clearer theoretical advance.

---

> ### Author Rebuttal · Authors · 2026-03-31
>
> Dear Reviewer,
>
> Thank you for the constructive feedback and for acknowledging the practical relevance of making OT solvers efficient on GPUs. We understand your concerns on theory, baselines, and originality, and clarify that our contribution is best viewed through **algorithm-system co-design**. (Added experiments, Figures R1-R6: https://1drv.ms/f/c/a685db1aaf460ebb/IgBatrSMd35xT5A_insTrqOEAV-dSSayYxWvy1n02-T0iTU)
>
> **1. Theory and dependence on $\eta$**
>
> We agree that making the dependence on $\eta$ explicit is essential if the goal is a tight global complexity bound. However, our paper is **not** positioned as providing a sharp, $\eta$-uniform iteration complexity for entropic-regularized OT. Rather, the theory targets a different (and necessary) objective:
> - **Convergence preservation under GPU-specific modifications:** cuRegOT introduces changes that alter the iterate sequence, so our main motivation is to show that we have **convergence preservation** guarantees under GPU-oriented modifications, rather than improving worst-case $\eta$-dependence. For transparency, we have provided an explicit form of the rate constant in the appendix:
> $$r = 1 - 2C\mu = 1 - \frac{2c\_{1}(1-c\_{2})\mu m^{2}\_{B}}{LM^{2}\_{B}}.$$
> In the revision we will explicitly state that $L/\mu$ and $M_B/m_B$ relate to conditioning and can implicitly depend on $\eta$, and we do not claim $\eta$-independent rates.
> - **Quasi-Newton theory is typically conservative:** Even for classical BFGS/L-BFGS, practical performance is often far better than worst-case, condition-number-based guarantees. Our goal follows this well-established paradigm: deliver **empirically efficient** solver on modern hardware, while maintaining rigorous convergence guarantees for the modified procedure.
> - **Worst-case safety via asynchronous Sinkhorn:** As described in Section 4.3, our algorithm compare quasi-Newton and Sinkhorn candidates and select the one with greater improvement. Thus, in the worst case, the method is bounded below by the Sinkhorn complexity. This block is modular and can be replaced by other GPU-friendly updates (including ones with stronger theoretical guarantees) without changing the overall framework.
> - **Evidence in the small-$\eta$ regime:** We add experiments at $\eta=0.01$ and $\eta=0.0001$ (Figures R2-R3). As $\eta$ decreases, all methods slow down due to ill-conditioning, but cuRegOT's advantage becomes more pronounced, indicating that incorporating curvature information is practically valuable in the hard regime.
>
> **2. Experimental scope and stronger alternatives**
>
> We agree that there exist accelerated methods with stronger rates than Sinkhorn. However, **theoretical superiority does not automatically translate to GPU wall-time gains**, and Sinkhorn (+Anderson) remains a strong baseline.
> - **Including Lin et al. (2022):** We add comparison with the accelerated Sinkhorn method of Lin et al. (2022) (Figure R5). Despite excellent iteration complexity, its GPU performance is not competitive in our tested settings, likely because it requires more gradient-like evaluations and extra passes per iteration, thus inflating constant factors.
> - **Why Sinkhorn (+Anderson) is a key baseline:** Vanilla and Anderson-accelerated Sinkhorn remain among the strongest and most widely-used GPU baselines due to their extreme parallelism and minimal per-iteration overhead. The fact that cuRegOT outperforms these baselines in wall-time supports our claim that it is competitive in realistic GPU regimes.
>
> We will include the new epxeriments into the final appendix.
>
> **3. Originality and algorithm-system co-design**
>
> We understand the concern about novelty and would like to emphasize that our contribution is not merely "porting SPLR to GPU".
> - **Not just implementation:** The key barrier for second-order methods on GPUs is sparse linear algebra, especially symbolic analysis and reordering for sparse Cholesky, which is difficult to parallelize and leads to GPU under-utilization. Addressing this is a genuine system challenge.
> - **Theory-guided system design:** Our designs are enabled and justified by theory: amortizing symbolic analysis relies on spectral properties of Hessian sparsification; asynchronous candidate insertion changes the optimization trajectory and requires a new convergence argument. These insights enable us to break the CPU-GPU synchronization bottleneck in a principled way.
> - **Practical impact:** Mainstream GPU OT libraries (e.g., POT & OTT-JAX) still largely rely on Sinkhorn and variants. cuRegOT offers a concrete path to bring quasi-Newton methods into GPU practice, particularly in high-accuracy regimes where first-order methods can be slow.
>
> We hope these clarifications, together with the new experiments and added baselines, address your concerns. We will revise the paper to better state the theoretical scope and to highlight the expanded experimental comparisons.

---

> > ### Author Rebuttal · Reviewer_PDeW · 2026-04-01
> >
> > I thank the authors for their reply. I understand that comparing their solver to publicly available libraries is a natural choice, and that, although many new algorithms have been proposed, existing libraries still rely on the same Sinkhorn-type algorithms.
> >
> > That said, I still think it would have been important to compare against other algorithms in order to justify the claim that their choice is empirically stronger.
> >
> > The unknown dependence on the regularization parameter, which appears to be exponential when the algorithm is analyzed through linear convergence rates, remains a concern for me.
> >
> > Given this, I maintain my original score, although I lower my confidence. In my view, this work is valuable, but I am not convinced it is a good fit for ICML. However, I may be mistaken.

---

> > > ### Author Response · Authors · 2026-04-07
> > >
> > > Dear Reviewer,
> > >
> > > We understand your concern on the linear rate that depends on $\eta$, and we would like to address this problem via the following arguments:
> > >
> > > 1. We **can** actually obtain a computational complexity that **does not** deteriorate exponentially with $\eta^{-1}$, although the convergence rate is sublinear instead of linear.
> > > 2. This is **consistent with the existing literature** on entropic-regularized OT: the Sinkhorn algorithm has two convergence rate upper bounds: a linear rate, but the constant is $1-e^{-O(\eta^{-1})}$ [1]; and a sublinear rate, which is a polynomial of $\eta^{-1}$ [2, 3].
> > > 3. Generally speaking, if one only considers the linear convergence rate, then it is almost certain that the constant is "bad" on $\eta^{-1}$; if one expects a polynomial dependence on $\eta^{-1}$, then the rate is almost certain to be sublinear.
> > >
> > > Below is the sketch of proof.
> > > 1. We can first assume that $\eta$ has an upper bound $\bar{\eta}$, since we only care about $\eta\rightarrow 0$.
> > > 2. Define the level set $$L_\eta(f_0)=\{x:f_\eta(x)\le f_0\},$$
> > > where $x=(\alpha,\beta_{-m})$ collects the free dual variables, and $f_0$ is a constant independent of $\eta$. We can prove that $L_\eta(f_0)$ must be contained in a box uniformly in $\eta$: for all $0<\eta\le\bar{\eta}$,$$L_\eta(f_0)\subseteq\{(\alpha,\beta_{-m}):\Vert\alpha\Vert_\infty\le U(f_0),\Vert\beta\Vert_\infty\le V(f_0)\}.$$
> > > 3. Assume $M_{ij}\ge 0$, and take initial value $x_0=\mathbf{0}$. Then $f(x_0)\le nm\bar{\eta}$, so $L_\eta(f(x_0))\subseteq L_\eta(nm\bar{\eta})$ must induce a uniform bound on the dual variables: $$\Vert\alpha\Vert_\infty\le U,\ \Vert\beta\Vert_\infty\le V,\ \text{for all }x\in L_\eta(f(x_0)).$$
> > > 4. The distance to the optimal point is bounded: for any $x\in L_\eta(f(x_0))$, $$\Vert x-x^\*\Vert^2=\Vert\alpha-\alpha^\*\Vert^2+\Vert\beta-\beta^*\Vert^2\le 4nU^2+4mV^2:=D^2.$$
> > > 5. We prove that $f(x)$ is $L$-smooth, where $L=O(\eta^{-2})$. This is because for $z=(u,v_{-m})$,$$z^T H(x) z=(1/\eta)\sum_{i,j}T_{ij}(x)(u_i+v_j)^2\le(2/\eta)S(x)\Vert z\Vert^2,$$ where $S(x)=\mathbf{1}\_n^T T(x)\mathbf{1}\_m$, and note that $\eta S(x)\le f\_\eta(x_0)+\alpha^T a+\beta^T b\le nm\bar{\eta}+U+V$. The rate can be improved to $O(\eta^{-1})$ as $x\rightarrow x^\*$, since $S(x^*)=1$.
> > > 6. For a general quasi-Newton method $x_{k+1}=x_k-\gamma_k B_k^{-1}g_k$ with $mI\preceq B_k\preceq MI$ and $\gamma_k$ satisfying the strong Wolfe condition, we have $$f(x_{k+1})\le f(x_k)-c_1(1-c_2)m/L\cdot\Vert\nabla f(x_k)\Vert^2_{B_k^{-1}}.$$ On the other hand, by convexity, $$f(x_k)-f^\*\le \Vert x_k-x^\*\Vert_{B_k}\cdot\Vert\nabla f(x_k)\Vert_{B_k^{-1}}.$$
> > > 7. Let $\Delta_k=f(x_k)-f^\*$. Since $\Vert x_k-x^\*\Vert_{B_k}\le MD$, we have $$\Delta_{k+1}\le\Delta_k-c_1(1-c_2)m/(LM^2D^2)\Delta_k^2.$$ Then by reduction, we get $$\Delta_k\le\frac{LM^2D^2}{c_1(1-c_2)m}\cdot k^{-1}.$$
> > >
> > > Overall, the optimization error is at the order of $O(\eta^{-2}/K)$, which is a quadratic function of $\eta^{-1}$.
> > >
> > > We hope this analysis addresses your concern on the theoretical property of the proposed solver, as the polynomial dependence on $\eta^{-1}$ matches the existing literature. And once again, we would like to highlight that our focus is on the algorithm-system co-design of a practical and efficient GPU solver for the entropic-regularized OT problems (note that our submission category is "**General Machine Learning->Hardware and Software**" instead of purely optimization theories).
> > >
> > > Also, in the rebuttal, we have already expanded the scope of our numerical experiments and have implemented a representative algorithm [4] on GPU that theoretically improves existing Sinkhorn-type methods.
> > >
> > > We are glad to provide additional information on the theoretical and experimental aspects of the proposed cuRegOT solver per request.
> > >
> > > [1] Carlier, G. (2022). On the linear convergence of the multimarginal Sinkhorn algorithm.
> > >
> > > [2] Altschuler, J., Niles-Weed, J., & Rigollet, P. (2017). Near-linear time approximation algorithms for optimal transport via Sinkhorn iteration.
> > >
> > > [3] Dvurechensky, P., Gasnikov, A., & Kroshnin, A. (2018). Computational optimal transport: Complexity by accelerated gradient descent is better than by Sinkhorn's algorithm.
> > >
> > > [4] Lin, T., Ho, N., & Jordan, M. I. (2022). On the efficiency of entropic regularized algorithms for optimal transport.

---

### Official Review · Reviewer_JnrR · 2026-03-11

**Soundness:** 3
**Presentation:** 4
**Significance:** 3
**Originality:** 3
**Overall Recommendation:** 5
**Confidence:** 3

**Summary:**

This paper proposes a GPU-accelerated Newton-based solver for discrete entropic-regularized optimal transport problems. The core idea is to 1) recycle the sparsity pattern of the Hessian across multiple iterations, 2) perform Sinkhorn iterations while the CPU works on reconstructs a new sparsity pattern, and 3) use a fused CUDA kernel for fast gradient computations. The proposed method has theory provided and experiments show the method is quite fast w.r.t. runtime.

**Compliance With Llm Reviewing Policy:**

Affirmed.

**Final Justification:**

Authors answered all my questions. My score remains.

**Key Questions For Authors:**

While some parts of this method are tailored to Entropic OT (e.g., the kernel), it seems that recycling sparsity patterns across iterations and using a more lightweight computation (e.g., gradient descent) could be used for more general optimization problems with sparse Hessians. What are the authors' thoughts about this?

**Limitations:**

The authors do not show/prove that the problem they are solving has slow-changing sparsity patterns of the Hessians, which their proposed methodology hinges on.

**Strengths And Weaknesses:**

Strengths

1. The paper gives good motivation for why the proposed method should be used
2. The method seems to be empirically strong for an important problem in machine learning
3. The proposed method seems to maximize available resources (GPU and CPU) in order to speed up the solver
4. Convergence is shown.

Weaknesses
1. The claim that ``We observe that the sparsity pattern, typically determined by the top-K entries in T , evolves slowly across iterations.'' is a compelling one. To that end, the paper would be much stronger if the authors could show this, theoretically and empirically. Theoretically, the authors could estimate a bound on how the sparsity patterns change using a sparsity measure (one idea is to use the Jaccard index), but there may be better options. Empirically, the authors should absolutely show the sparsity patterns of the Hessian across a few iterations in the experimental section.
2. While a description of how $H_\Omega$ is offloaded to prior works, a brief description of how it is computed could make this paper more self-contained and easier for the reader.

Minor typos:
1. Line 175: "may sparser" should be "may be sparser"
2. Line 197: "Below provides" should be "Below we provide"

---

> ### Author Rebuttal · Authors · 2026-03-31
>
> Dear Reviewer,
>
> Thank you for the thoughtful feedback and concrete suggestions. Below we clarify what we can support now and what we will add in the revision. (Added experiments, Figures R1-R6: https://1drv.ms/f/c/a685db1aaf460ebb/IgBatrSMd35xT5A_insTrqOEAV-dSSayYxWvy1n02-T0iTU)
>
> **1. Theory and empirical evidence for the slow evolution of sparsity patterns**
>
> In cuRegOT, we simply build the sparsification set $\Omega$ from the top-K entries of the current transport plan matrix $T(x)$, and we state the empirical observation that this pattern evolves slowly, which motivates reusing the symbolic analysis result for $S$ iterations.
>
> **A simple theoretical intuition**
>
> The matrix entries satisfy $T_{ij}=\exp((\alpha_i+\beta_j-M_{ij})/\eta)$.
> Therefore, between two consecutive iterates $x$ and $x^{+}$,
> $$
> \frac{T_{ij}(x^{+})}{T_{ij}(x)}=\exp\left(\frac{\Delta\alpha_i+\Delta\beta_j}{\eta}\right).
> $$
> So if $\Vert\Delta\alpha\Vert_\infty+\Vert\Delta\beta\Vert_\infty$ is small relative to $\eta$ (which is typical once line search stabilizes), relative changes in $T$ are controlled. Moreover, the top-K set would be invariant whenever the margin between the $K$-th and $(K{+}1)$-th entries (in logarithmic domain) exceeds the maximum perturbation induced by $\Delta\alpha,\Delta\beta$. In the revision we will add this intuition.
>
> **Supporting evidence from OT sparsity structure**
>
> For unregularized OT, transport plans are known to be sparse (at most $(n+m-1)$ nonzeros, Proposition 3.4 of [1]). Additionally, recent results on "rigidity" show that for generic costs, each source splits its mass only across a narrow band of targets and the support has strong structural constraints [2]. Since entropic-regularized OT approximates unregularized OT as $\eta\to 0$, this supports the idea that the mass in $T$ concentrates and the effective support is structured, making the top-K support less volatile across iterations.
>
> **Empirical visualization**
>
> We have added a new figure (Figure R6) showing the evolution of the transport plan across iterations (heatmaps over multiple iterates). The changes are gradual rather than abrupt. This visualization again supports the "slowly changing pattern" premise.
>
> We will add these figures and discussions in the revised manuscript.
>
> **2. Making $H_{\Omega}$ construction self-contained**
>
> We agree that a brief description is helpful. In our implementation, on iterations where we refresh the pattern, we use a simple top-K rule: select the largest $K$ entries of $T$, forming $T^{\Omega}$ by zeroing out all other entries. Then $H_{\Omega}$ is assembled using the formula given in Section 3.1:
> $$
> H_{\Omega}=\eta^{-1}\begin{bmatrix}\mathbf{diag}(T\mathbf{1}\_{m}) & T^{\Omega}\_{-m}\\\\
> (T^{\Omega}\_{-m})^{T} & \mathbf{diag}(T^{T}\_{-m}\mathbf{1}\_{n})
> \end{bmatrix}.
> $$
>
> We will add a short paragraph summarizing these steps in the revised manuscript.
>
> **3. Generality beyond entropic OT**
>
> We agree that this is an excellent direction. The two scheduling ideas—(1) recycling sparsity patterns and (2) doing lightweight progress while CPU handles symbolic work—are broadly applicable. The key challenges, as you point out, are not the ideas themselves but the guarantees:
>
> - **Positive definiteness preservation after sparsification:** In entropic-regularized OT we can leverage the problem structure to ensure that the (quasi-)Newton system remains invertible and well-posed.
> - **Convergence under changing patterns/candidate insertion:** Reusing a pattern and inserting auxiliary steps changes the optimization trajectory, so standard quasi-Newton proofs do not automatically apply; this is why we provide a dedicated convergence argument for our modified procedure.
>
> For more general sparse-Hessian problems, one would need analogous conditions (e.g., uniform spectral bounds for the approximate Hessian, or a trust-region/line-search mechanism plus damping) and a "safe insertion" criterion (monotone decrease or sufficient decrease). We view this as promising future work.
>
> We have also corrected the typos, and we sincerely thank the reviewer for pointing out them.
>
> [1] Peyré, G., & Cuturi, M. (2019). Computational optimal transport: With applications to data science.
>
> [2] Johnson, A. B., & Steinerberger, S. (2023). Rigidity of Kantorovich solutions in discrete Optimal Transport. arXiv preprint arXiv:2311.18764.

---

> > ### Author Rebuttal · Reviewer_JnrR · 2026-04-01
> >
> > All of my questions have been resolved. I maintain my score of Accept.

---

> > > ### Author Response · Authors · 2026-04-03
> > >
> > > Thank you for the positive comments and insightful suggestions!

---

### Official Review · Reviewer_PkPW · 2026-03-11

**Soundness:** 4
**Presentation:** 4
**Significance:** 3
**Originality:** 4
**Overall Recommendation:** 5
**Confidence:** 4

**Summary:**

The paper proposes cuRegOT, a high-performance GPU solver for entropic-regularized optimal transport (OT) via SPLR quasi-Newton method with special designs for modern GPUs. The authors introduce three key techniques: (1) amortized symbolic analysis that reuses sparsity patterns across iterations, (2) collaborative CPU-GPU computation where asynchronous Sinkhorn iterations generate candidate solutions while the CPU performs symbolic analysis, and (3) a fused CUDA kernel that computes transport plans and gradients in a single bandwidth-efficient pass. The paper provides theoretical guarantees (global convergence and at least linear rate) for the modified algorithm and presents extensive synthetic and CIFAR-10 experiments showing substantial speedups over GPU Sinkhorn and Anderson-accelerated variants, especially at higher accuracies and larger scales.

**Compliance With Llm Reviewing Policy:**

Affirmed.

**Key Questions For Authors:**

1. How sensitive is cuRegOT to the choice of hyperparameters (e.g., the number of iterations) Could you provide additional ablations showing the trade-off between CPU overhead, GPU utilization, and convergence speed as these parameters vary?
2. Have you evaluated cuRegOT on highly ill-conditioned cost structures, where Hessians may be more challenging? How does the method compare to Sinkhorn and other solvers in terms of numerical stability and required precision in these regimes?
3. What would be required to integrate cuRegOT into existing OT libraries and use it as a backend?

**Limitations:**

Further validation on a broader range of real-world applications (e.g., domain adaptation, generative models) and on diverse hardware/backends would strengthen the case for widespread adoption. Additionally, more systematic study of memory usage, hyperparameter sensitivity, and behavior in low-regularization or near-unregularized regimes would clarify the robustness boundaries of cuRegOT.

**Strengths And Weaknesses:**

Strengths
Clear identification of a real bottleneck: The paper precisely pinpoints the CPU-side symbolic analysis in sparse Cholesky as the limiting factor for GPU-accelerated Newton-type OT methods and designs solutions that directly targets this issue.
Well-motivated algorithmic design choices: The combination of amortized symbolic analysis, asynchronous candidate iterations via Sinkhorn, and fused gradient kernel are coherent and well grounded in modern GPU architecture considerations.
Scalability emphasis: The benefits of cuRegOT are more pronounced for larger problem sizes and for when a relatively high level of precision is required, which are precisely the areas where regularized OT is currently a limitation in practice.


Weaknesses
Limited diversity of real-world tasks: All real data experiments are based on CIFAR-10 feature-based OT; there is no demonstration on other modalities or more application-driven OT use cases, which would better showcase practical impact.
Memory footprint and numerical robustness: The paper focuses on runtime but gives relatively little insight into peak memory usage, behavior under extreme sparsity/ill-conditioning and overall issues that can be critical in large-scale OT deployments.
Interface with higher-level libraries: It remains somewhat unclear how easily cuRegOT can be integrated into popular OT/tooling ecosystems and what API assumptions or constraints are imposed.

---

> ### Author Rebuttal · Authors · 2026-03-31
>
> Dear Reviewer,
>
> Thank you for the positive assessment and for the detailed, practical questions. Below are our point-to-point responses. (Added experiments, Figures R1-R6: https://1drv.ms/f/c/a685db1aaf460ebb/IgBatrSMd35xT5A_insTrqOEAV-dSSayYxWvy1n02-T0iTU)
>
> **1. Diversity of real-world tasks**
>
> We agree that broader application-driven evaluations would strengthen impact. Our current goal is to **improve the core computational efficiency** for entropic-regularized OT—given $M,a,b,\eta$—rather than to propose a new OT-based machine learning model. This is why we focused on extracting cost matrices from standard benchmarks (synthetic + CIFAR-10 feature OT) and studying the accuracy-time curves in a controlled setting.
>
> In future work, we plan to embed cuRegOT as a solver backend in typical OT pipelines such as domain adaptation and generative modeling (where OT is repeatedly solved inside training loops).
>
> **2. Memory footprint and numerical robustness**
>
> **Memory:** We did not include peak GPU memory comparisons because different frameworks use different allocators and memory pools (e.g., JAX preallocates 75% of the memory, PyTorch caches freed memory, and CuPy also maintains memory pools), making direct cross-framework reporting difficult to standardize. In future work, we plan to explore the use of lower-level tools to report memory under a unified measurement protocol. It is worth mentioning that our fused-kernel design is explicitly motivated to reduce memory traffic by computing elements and gradient reductions in a single pass.
>
> **Robustness/ill-conditioning:** We have added experiments in harder regimes by considering more $\eta$ values (Figures R2, R3), where smaller $\eta$ leads to more ill-conditioning. The results show that while Sinkhorn and its variants deteriorate rapidly at smaller $\eta$ values, cuRegOT maintains a fast convergence speed. This indicates that the second-order information we utilize is practically indispensable for overcoming the ill-conditioning caused by small $\eta$.
>
> **3. Integration with higher-level libraries**
>
> We have implemented NumPy and PyTorch interfaces. When the input is a CUDA tensor, the PyTorch interface supports zero-copy invocation (no host-device transfers), so cuRegOT can be used inside GPU training loops as a drop-in solver once $M,a,b,\eta$ are available. In fact, the current experiments are conducted using the PyTorch interface. We have planned to open-source cuRegOT in the future, so that it can be easily integrated into existing OT ecosystems.
>
> **4. Hyperparameter sensitivity**
>
> The main hyperparameter controlling amortization is $S$, the number of iterations for which we reuse the sparsity pattern (and thus reuse symbolic analysis). We have added an explicit $S$-sensitivity study (Figure R4) to answer this. We find that the method is relatively robust: $S\in[5, 30]$ consistently yields good accuracy-time curves across problems, while $S=1$ (no reuse) can be slower due to CPU overhead.
>
> We appreciate the suggestions and will strengthen the paper by including the additional experiments and discussions.

---

> > ### Author Rebuttal · Reviewer_PkPW · 2026-04-03
> >
> > Thank authors for the reply. The authors have addressed most of my concerns.

---

> > > ### Author Response · Authors · 2026-04-03
> > >
> > > Thank you for the positive feedback and constructive suggestions!

---

### Official Review · Reviewer_GHLs · 2026-03-11

**Soundness:** 3
**Presentation:** 3
**Significance:** 2
**Originality:** 2
**Overall Recommendation:** 4
**Confidence:** 3

**Summary:**

The paper is dedicated to solving large-scale entropy-regularized discrete optimal transport. The respective dual objective possesses one redundant degree of freedom, which can be eliminated by fixing one dual variable component. This ensures strict convexity of the objective. A recently proposed sparse-plus-low-rank (SPLR) algorithm is a quasi-Newton method designed for efficient minimization of this objective. Its use in large-scale GPU-oriented scenarios is hindered by the fact that computation of the Hessian’s sparsification scheme is a CPU-heavy operation.

The present work proposes a few modifications to the method that make it GPU-friendly. First, the authors argue that the sparsification scheme should only be computed from time to time, and reused in a few subsequent iterations. Second, when the sparsification scheme is being recomputed on the CPU, the modified method also performs a few Sinkhorn steps on the (otherwise idle) GPU. If these steps result in a point with a lower objective value than that of the new SPRL iterate, the method continues its operation from the point generated by the Sinkhorn steps. Finally, the work develops optimizations for CUDA-efficient gradient evaluation. Empirical comparison with GPU-oriented solvers is presented.

**Compliance With Llm Reviewing Policy:**

Affirmed.

**Final Justification:**

The rebuttal addressed most of my concerns. However, the issue of limited novelty remains, as the modifications are mainly engineering improvements rather than significant theoretical contributions. I raise my score from 2 to 4 (weak accept) as the paper looks practically useful.

**Key Questions For Authors:**

1. Which value of $S$ (the number of iterations with the same sparsity pattern) is used in experiments? How does changing $S$ affect empirical performance?
1. Could you conduct experiments with other values of the entropy regularization coefficient?
1. Could you also empirically compare your method to accelerated Sinkhorn from the paper “On the efficiency of entropic regularized algorithms for optimal transport” by T. Lin et al.?
1. I haven't found the code of the experiments. Did you include it in the submission?

Minor remarks:
1. In line 137, should $b$ be replaced with $b_{-m}$ in the formula of the gradient?
1. Line 175, left column: "may sparser" -> "may be sparser"
1. Please update the running title (currently it's "Submission and Formatting Instructions for ICML 2026")

**Limitations:**

Yes

**Strengths And Weaknesses:**

The paper's strengths are its clear structure and the focus on a relevant and well-motivated topic.

Weaknesses:
1. The modification of SPLR based on reusing sparsity patterns doesn’t constitute a notable contribution. Indeed, the original SPLR paper already established that using only the index set $\Omega^{\star}$ is sufficient for convergence, if I understand correctly. The authors don’t explore this aspect further, e.g., by theoretically investigating how $S$ (the number of iterations with the same sparsity pattern) affects the convergence rate.
1. The modification based on introducing Sinkhorn steps is also not a substantial theoretical contribution; it is justified by a relatively simple argument.
1. The performance metric in the experiments is not very informative. According to lines 308-310 (right column), the authors define “optimization error” as the marginal error, i.e., constraint violation. If my understanding is correct, they don’t show the optimality gap in the experiments. Thus, if a method simply returned the product measure $ab^T$, it would instantly achieve zero “optimization error”.
1. The experiments are limited to a single value of the entropy regularization coefficient.

---

> ### Author Rebuttal · Authors · 2026-03-31
>
> Dear Reviewer,
>
> We sincerely thank you for your constructive feedback and for recognizing the practical relevance of our work in solving large-scale OT on GPUs. Below are our point-to-point responses. (Added experiments, Figures R1-R6: https://1drv.ms/f/c/a685db1aaf460ebb/IgBatrSMd35xT5A_insTrqOEAV-dSSayYxWvy1n02-T0iTU)
>
> **1. On the role of Theorem 4.1 and algorithm-system co-design (Weakness 1)**
>
> We completely agree that the theoretical foundation for reusing the sparsity pattern was established in the SPLR paper, and we have explicitly credited this in the text. However, we would like to highlight the difference in how this theorem is utilized between the two works:
>
> - **SPLR perspective: retrospective justification**: The original SPLR algorithm was designed for CPU execution, where symbolic analysis is not a bottleneck compared to dense gradient computations. Consequently, it is natural to update the pattern in every iteration (e.g., top-K) to better approximate the Hessian, and the theorem serves as a theoretical guarantee that this heuristic is safe.
> - **cuRegOT perspective: proactive system design**: In the GPU regime, the serial nature of symbolic analysis becomes the primary bottleneck. Our key contribution is an algorithm-system co-design that uses Theorem 4.1 as an enabling principle: we can intentionally choose a scheme that makes GPU execution feasible, and the theorem acts as the catalyst for a hardware-specific system design that breaks the frequent and costly CPU-GPU synchronization barrier.
>
> **2. The necessity and non-triviality of the convergence proof (Weakness 2)**
>
> We agree that the acceptance logic ("keep the candidate with lower objective") is intentionally simple. The novelty is not in inventing a complicated criterion, but in turning unavoidable CPU bottleneck into useful GPU progress via asynchronous candidate generation. Its theoretical integration is also not trivial:
>
> Inserting auxiliary iterates infinitely often can break standard quasi-Newton proofs (dependency between successive iterates), so SPLR's original convergence argument does not directly apply to our modified trajectory. We therefore provide a new convergence analysis for the modified algorithm. Therefore, while the acceptance rule is simple, the result is a non-trivial "safe insertion" principle that justifies the asynchronous CPU–GPU design.
>
> Regarding your concerns about the theoretical contributions, we would like to clarify that the core positioning of our work is not pure optimization theories, but rather an algorithm-system co-design.
>
> **3. Experimental scope (Weaknesses 1 & 4, Questions 1-3)**
>
> You correctly pointed out the need to study how $S$ affects solver's performance (we globally used $S=10$ in the original experiments). Since deriving tight theoretical bounds for $S$ is notoriously difficult, we have added an explicit $S$-sensitivity study (Figure R4) to answer this. We find that the method is relatively robust: $S\in[5, 30]$ consistently yields good accuracy-time curves across problems, while $S=1$ (no reuse) can be slower due to CPU overhead.
>
> To test solver's performance under different regularization parameters, we have added experiments evaluating $\eta=0.01$ and $\eta=0.0001$ (Figures R2, R3). The results show that while Sinkhorn and its variants deteriorate rapidly at smaller $\eta$ values, cuRegOT maintains a fast convergence speed. This indicates that the second-order information we utilize is practically indispensable for overcoming the ill-conditioning caused by small $\eta$.
>
> In addition, we have added comparison with the accelerated Sinkhorn algorithm proposed by Lin et al. (2022). Although this method has an outstanding theoretical convergence rate, in our experiments, its actual GPU performance is not very competitive (Figure R5). A major reason for this is that such acceleration requires more objective/gradient-like evaluations or extra passes per iteration, significantly inflating the constant factors in its computational complexity.
>
> We will include these new experiments and discussions in the revised manuscript.
>
> **4. Performance metric (Weakness 3)**
>
> This is a good point. For general algorithms, only testing the marginal error is not sufficient. What is special here is that the methods compared in this paper are based on the dual problem, meaning that the output transport plan is obtained as $T_{ij}=\exp((\alpha_i+\beta_j-M_{ij})/\eta)$. Under this special structure, the marginal error is exactly the gradient norm of the dual problem. Therefore, testing the gradient is a valid indicator of the optimality.
>
> **5. Code availability (Question 4)**
>
> We have planned to release the code after the overall method receives sufficient reviews. In case the code availability is your concern, we can upload the core CUDA kernels to the link above, if this is permitted.
>
> We have also corrected the typos and other formatting issues, and we sincerely thank the reviewer for pointing out them.

---

> > ### Author Rebuttal · Reviewer_GHLs · 2026-04-01
> >
> > Dear authors, thank you for the responses. Before I adjust the score, could you please plot the duality gap curves of cuRegOT and other solvers, at least in some experiments? I think this is important to understand how well the algorithm approximates the OT distance. Although the duality gap is closely related to constraint violation, it can't be calculated solely based on the value of marginal error, as far as I know.

---

> > > ### Author Response · Authors · 2026-04-03
> > >
> > > Sure. We have uploaded **Figures R7-R8** to https://drive.google.com/drive/folders/1Uu74ubxxvvyo8puLnewDBcrYAVtbNO99 (the previous link is somehow broken for now, so we change the link), which show both the marginal error and duality gap against wall-time. We also include **Figures R9-R10** to study the ill-conditioned case of $\eta=0.0001$. All these figures include the accelerated Sinkhorn algorithm proposed by Lin et al. (2022).
> > >
> > > The plots show that the duality gap demonstrates very similar patterns as the marginal error. Inspired by your comment, we have discovered an interesting relation between the duality gap and the marginal error for the entropic-regularized OT. By definition, the duality gap is the difference between the current primal value $L_p$ and the current dual value $L_d$, where
> > > $$
> > > \begin{align*}
> > > L_{p} & =\langle T,M\rangle-\eta\cdot h(T)=\langle T,M\rangle+\eta\cdot\sum_{i,j}T_{ij}\cdot\log(T_{ij})-\eta\cdot\sum_{i,j}T_{ij},\\\\
> > > L_{d} & =-\eta\cdot\sum_{ij}T_{ij}+\alpha^{T}a+\beta^{T}b,
> > > \end{align*}
> > > $$
> > > and $T_{ij}=\exp\{\eta^{-1}(\alpha_{i}+\beta_{j}-M_{ij})\}$. After some simplification, we can obtain that
> > > $$
> > > \begin{align*}
> > > L_{p}-L_{d} & =\langle T,M\rangle+\sum_{i,j}T_{ij}(\alpha_{i}+\beta_{j}-M_{ij})-\alpha^{T}a-\beta^{T}b\\\\
> > >  & =\sum_{i,j}T_{ij}(\alpha_{i}+\beta_{j})-\alpha^{T}a-\beta^{T}b\\\\
> > >  & =\alpha^{T}(r-a)+\beta^{T}(c-b),
> > > \end{align*}
> > > $$
> > > where $r$ is the row sum vector of $T$ and $c$ is the column sum vector. Note that $\Vert r-a\Vert_1+\Vert c-b\Vert_1$ is the marginal error, so we have
> > > $$
> > > \begin{align*}
> > > |L_{p}-L_{d}| & \le\Vert\alpha\Vert_{\infty}\Vert r-a\Vert_{1}+\Vert\beta\Vert_{\infty}\Vert c-b\Vert_{1}\\\\
> > >  & \le\max\\{\Vert\alpha\Vert_{\infty},\Vert\beta\Vert_{\infty}\\}\cdot(\Vert r-a\Vert_{1}+\Vert c-b\Vert_{1}).
> > > \end{align*}
> > > $$
> > > In some sense (not very strict since there is a factor about the norms of $\alpha$ and $\beta$), the marginal error provides an upper bound for the duality gap. This may partly explain why the two metrics show very similar patterns. As a whole, we completely agree that reporting both the marginal error and the duality gap provides a more comprehensive characterization of the convergence behavior of solvers.
> > >
> > > As a final note, we would like to once again highlight the aim of this paper:
> > >
> > > 1. Our focus is on the algorithm-system co-design of a practical and efficient GPU solver for the entropic-regularized OT problems (note that our submission category is "**General Machine Learning->Hardware and Software**" instead of purely optimization theories). **Figure R1** shows the pipeline of cuRegOT that highlights our contributions.
> > > 2. Our theory, which does **not** automatically hold from existing literature, validates that the algorithm **preserves** a global linear convergence guarantee **under GPU-oriented modifications**.
> > > 3. Per the suggestions by all the reviewers, we have **expanded the scope** of our numerical experiments, and have demonstrated the performance of cuRegOT from more different angles. We have shown that it is consistently competitive among recent and advanced algorithms, is robust to hyperparameters, and remains effective under ill-conditioned problems.
> > >
> > > We are glad to provide additional information on the theoretical and experimental aspects of the proposed cuRegOT solver per request.

---

### Decision · Program_Chairs · 2026-04-30

**Decision:**

Accept (regular)

**Comment:**

This paper presents **cuRegOT**, a GPU-oriented solver for entropic-regularized optimal transport based on a sparse-plus-low-rank quasi-Newton method together with several system-level modifications. The paper targets a practically important bottleneck in large-scale OT, namely that existing GPU-friendly methods such as Sinkhorn can be slow in difficult regimes, while stronger second-order methods are often poorly matched to GPU hardware.

The reviewers were broadly positive about the paper’s practical significance and systems contribution. Several reviewers also viewed the convergence analysis as important because the proposed CPU–GPU modifications alter the original optimization trajectory and therefore require nontrivial justification.

The main concerns raised in review were about novelty and evaluation scope, such as some of the individual modifications are more engineering-driven than theoretically novel, stronger comparisons beyond Sinkhorn-style baselines are needed, as well as more explicit discussion of the dependence on the regularization parameter. These are fair concerns, which are fairly addressed during the rebuttal phase. It is encourged the authors properly incorporate the rebuttal into the final version of the paper.